# Term2Note: Synthesising Differentially Private Clinical Notes from Medical Terms

## Abstract

Training data is fundamental to the success of modern machine learning models, yet in high-stakes domains such as healthcare, the use of real-world training data is severely constrained by concerns over privacy leakage. A promising solution to this challenge is the use of differentially private (DP) synthetic data, which offers formal privacy guarantees while maintaining data utility. However, striking the right balance between privacy protection and utility remains challenging in clinical note synthesis, given its domain specificity and the complexity of long-form text generation. In this paper, we present **Term2Note**, a methodology to synthesise full-length clinical notes under strong DP constraints. By structurally separating content and form, Term2Note generates section-wise note content conditioned on medical terms, with terms and notes privatised under separate DP constraints. A DP quality maximiser further enhances synthetic notes by selecting high-quality outputs. Experimental results show that Term2Note produces synthetic notes with statistical properties closely aligned with real clinical notes, demonstrating strong fidelity. In addition, multi-label classification models trained on these synthetic notes perform comparably to those trained on real data, confirming their high utility. Compared to existing DP text generation baselines, Term2Note achieves substantial improvements in both fidelity and utility, while avoiding reliance on label distribution assumptions, suggesting its potential as a viable privacy-preserving alternative to using sensitive clinical notes.

## 1 Introduction

The scaling law of neural language models (Kaplan et al., 2020) suggests that model performance improves substantially with increased dataset size, i.e., larger training corpora generally lead to lower test loss. As large language models (LLMs) continue to scale in size, with models such as Llama 3 (Meta, 2024), Gemma 3 (Google, 2025), and Qwen3 (Yang et al., 2025) ranging from 0.6B to over 405B parameters, the demand for large-scale, high-quality training data has risen accordingly. To meet this demand, synthetic data generation using LLMs has emerged as a promising direction. Instruction-following synthetic datasets (Schick & Schütze, 2021; Taori et al., 2023; Li et al., 2025b) have demonstrated impressive effectiveness for model pretraining and fine-tuning. This is especially relevant for high-stakes domains such as healthcare (Li et al., 2025a), where real data is often siloed, heavily regulated, and difficult to share (Schlegel et al., 2025). Although large amounts of clinical data exist within healthcare institutions, access to these datasets remains extremely limited due to their sensitive nature and the strict privacy regulations surrounding them. A practical and privacy-conscious solution is to share synthetic versions of sensitive clinical data instead of the raw data itself. However, to make such synthetic sharing viable, formal privacy guarantees are essential.

Differential privacy (DP) provides a principled framework for this purpose (Alzoubi & Mishra, 2025). By bounding the influence of any individual record on the synthesised dataset, DP offers quantifiable privacy guarantees. Prior work on DP-based text generation has focused mainly on short-form texts in low-risk domains such as reviews (Yue et al., 2023; Kurakin et al., 2023; Mattern et al., 2022; Flemings & Annavaram, 2024). In biomedical settings, efforts have been restricted to synthesising (public) PubMed abstracts (Xie et al., 2024) and relatively short clinical passages (Aziz et al., 2022; Ramesh et al., 2024), with no prior work addressing the more complex task of generating full-length clinical notes under DP constraints. Synthesising clinical notes with DP presents two key challenges. First, *generation complexity* (Kweon et al., 2024a; Weetman et al., 2021): clinical

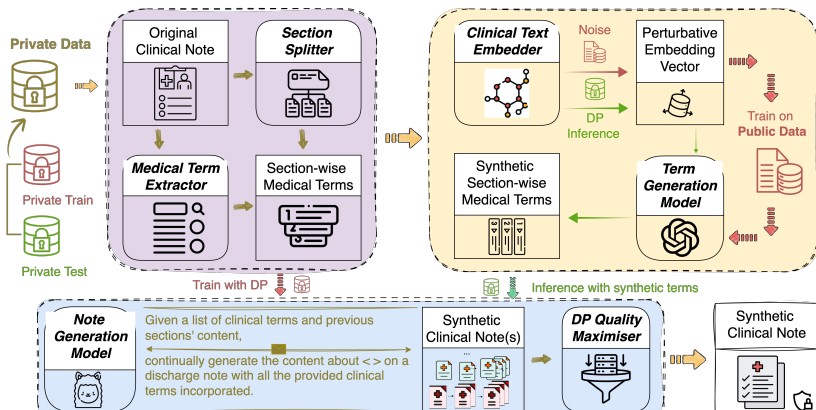

Figure 1: Overview of **Term2Note**. An original clinical note is split into sections and associated medical terms, which are embedded and optionally privatised via DP perturbation. A term generation model produces synthetic terms from these embeddings, and a DP-trained note generation model synthesises section-wise notes conditioned on them. A DP quality maximiser then selects the final synthetic note. Appendix K illustrates the whole process with a concrete example.

notes are long and exhibit diverse structures and free-form content, making it more difficult for generative models to maintain coherence and quality, particularly under privacy constraints. Second, *domain specificity* (Adnan et al., 2010): clinical notes typically contain extensive domain-specific terminologies, which require expert knowledge to understand and reproduce accurately.

In this paper, we tackle this underexplored and challenging task by proposing Term2Note, a novel methodology for DP synthetic clinical note generation. As illustrated in Figure 1, Term2Note addresses the challenge of long-form generation by leveraging domain-specific document structures to decompose the task into smaller, section-wise subtasks. To handle domain specificity, it conditions generation not on generic metadata (e.g., class labels or diagnosis codes such as *Enterocolitis due to Clostridium difficile*), but on salient clinical terms (e.g., *[diarrhea, Clostridium difficile colitis, Vancomycin]*), which are subjected to an additional layer of DP protection. This term-based conditioning strategy allows the model to generate text that is both clinically meaningful and structurally coherent, while also enabling more fine-grained privacy control. Our experimental results show that Term2Note: (1) produces synthetic notes with high structural and semantic fidelity to real clinical data; (2) enables strong utility in the downstream task, such as ICD code prediction; (3) satisfies formal DP guarantees that make it suitable for safe data sharing. Moreover, Term2Note consistently outperforms baseline methods across all evaluation metrics, often by a large margin. In summary, Term2Note offers a promising solution for privacy-preserved clinical notes sharing.

## 2 BACKGROUND & RELATED WORK

**Definition 1.** (Dwork et al., 2006) A randomised algorithm $\mathcal{M} : \mathbb{N}^{|\mathcal{X}|} \to R$ is said to be $(\epsilon, \delta)$-**differentially private (DP)**, if, for any two neighboring datasets $D$ and $D'$ differing in one single instance, and for all subsets $S$ of the output space of $\mathcal{M}$, it has $\mathbb{P}[\mathcal{M}(D) \in S] \le e^\epsilon \mathbb{P}[\mathcal{M}(D') \in S] + \delta$.

The definition implies the probability distributions induced by $\mathcal{M}$ on neighboring datasets must be close, with their likelihood ratios bounded by a multiplicative factor of $e^\epsilon$ and an additive slack of $\delta$. Smaller $\epsilon$ values indicate stronger privacy, while $\delta$ denotes the (typically negligible) probability of a privacy breach exceeding the $e^\epsilon$ bound.

**Theorem 1.** (**Post-Processing**) (Dwork & Roth, 2014) Let $\mathcal{M} : \mathbb{N}^{|\mathcal{X}|} \to R$ be a randomised algorithm that is $(\epsilon, \delta)$-DP. Let $f : R \to R'$ be an arbitrary randomised function. Then $f \circ \mathcal{M} : \mathbb{N}^{|\mathcal{X}|} \to R'$ is also $(\epsilon, \delta)$-DP.

**Theorem 2.** (**Parallel Composition**) (McSherry, 2009) Let dataset $D = D_1 \cup \cdots \cup D_k$, and $D_i \cap D_j = \emptyset$ for $i \ne j$. Let $\mathcal{M}_i : \mathbb{N}^{|\mathcal{X}_i|} \to R_i$ be a randomised algorithm that is $(\epsilon_i, \delta_i)$-DP, for $i \in [k]$. Then $\mathcal{M}(D) = (\mathcal{M}_1(D_1), \cdots, \mathcal{M}_k(D_k))$ is $(\max_i \epsilon_i, \max_i \delta_i)$-DP.

The post-processing property ensures that once a randomised algorithm $\mathcal{M}$ satisfies $(\epsilon, \delta)$-DP, any deterministic or randomised function applied to its output cannot weaken its privacy guarantee. Thus, a generative model trained under DP retains its privacy during downstream use, e.g., generating synthetic data, applying further transformations, or training downstream models. The parallel composition property states that applying DP mechanisms to disjoint data subsets yields an overall privacy loss bounded by the maximum individual $(\epsilon, \delta)$ values—a key feature in our method, where clinical terms and notes can be privatised independently.

**DP in Deep Learning** can be achieved by injecting random noise into the input data, where the noise is typically sampled from a pre-determined distribution, such as the Gaussian distribution. In the context of deep learning, the DP-SGD algorithm (Abadi et al., 2016) introduces a principled way to achieve this by clipping per-sample gradients and adding noise at each optimisation step. This approach ensures that the influence of any single training example on the model's parameters remains bounded, thereby enforcing DP guarantees throughout training. Some subsequent research (Bu et al., 2023; Yousefpour et al., 2021; Lee & Kifer, 2020) has focused on improving the computational efficiency of DP-SGD, aiming to reduce the time and memory overhead associated with per-example gradient computation. Unless otherwise specified, we adopt the FastDP algorithm (Bu et al., 2023) to fully fine-tune the model under DP constraints throughout this paper.

**Synthetic text generation** has progressed rapidly with the rise of LLMs, especially through instruction-following datasets that enhance downstream performance (Taori et al., 2023; Peng et al., 2023; Schick & Schütze, 2021). Generating synthetic clinical text, however, is substantially more challenging due to the domain's complexity, reliance on expert knowledge, and lack of easily templated instructions. Prior attempts have often produced datasets with limited downstream utlity (Li et al., 2023a; Schlegel et al., 2023). Recent work has sought to improve realism by prompting LLMs with control code (e.g., ICD codes) (Falis et al., 2024) or transforming biomedical abstracts (e.g., PubMed content) into clinical-style text (Kweon et al., 2024b). While promising, these methods do not address privacy, a central concern in clinical settings. A straightforward strategy is de-identifying private information and prompting LLMs to fill in the gaps (Sarkar et al., 2025), but a more principled approach is to fine-tune or instruction-tune generative models with DP for formal protection (Aziz et al., 2022; Ramesh et al., 2024; Baumel et al., 2024). However, prior studies operate on relatively short clinical texts, either naturally brief or deliberately truncated, making generation considerably easier. In contrast, DP-constrained generation of full-length clinical notes, as addressed in this paper, is more challenging due to their length, unstructured format, and high variability.

## 3 METHODOLOGY

### 3.1 PROBLEM STATEMENT

Given a private dataset $D^{\mathrm{src}}$ consisting of clinical notes, our goal is to develop a mechanism $\mathcal{M}$ that satisfies $(\epsilon, \delta)$-DP and produces a synthetic dataset $D^{\mathrm{syn}}$ as its output. Let $X^{\mathrm{src}} \in D^{\mathrm{src}}$ denote an original clinical note and $X^{\mathrm{syn}} \in D^{\mathrm{syn}}$ its corresponding synthetic version. To support the development and evaluation of the mechanism, we partition the private dataset into a training set $D^{\mathrm{src}}_{\mathrm{train}}$ and a test set $D^{\mathrm{src}}_{\mathrm{test}}$. The training set is used to develop $\mathcal{M}$, while the test set remains completely unseen by $\mathcal{M}$ to provide an unbiased evaluation. Additionally, we assume access to a public dataset of clinical terms, denoted as $D_{\mathrm{public}}$, which can be automatically derived from publicly available medical resources and therefore used freely without privacy constraints.

We introduce **Term2Note**, a section-wise DP generation framework for clinical notes. An overview is illustrated in Figure 1, with the detailed procedure in Algorithm 1. In the following, we first elaborate on Term2Note alongside the algorithm, and then present the implementation details.

### 3.2 FORMAT AND TERM IDENTIFICATION

Since our framework synthesises clinical notes via section-wise generation conditioned on medical terms, we first standardise the structure of the notes and identify salient clinical terms. Let SECSPLIT denote an automatic section segmentation module. Given an original clinical note $X^{\mathrm{src}}$, SECSPLIT outputs a list of $m$ segmented sections: $[\mathrm{SEC}^{\mathrm{src}}_1, ..., \mathrm{SEC}^{\mathrm{src}}_m] = \mathrm{SECSPLIT}(X^{\mathrm{src}})$, where $\mathrm{SEC}^{\mathrm{src}}_i = $ "" if the $i$-th section is absent. Next, we apply an automatic term extraction module, denoted as

---

**Algorithm 1** Term2Note

---

**Input:** Private note $X^{\text{src}}$, public term lists $D_{\text{public}}$, privacy parameters for term generation $(\epsilon_t, \delta_t)$ and note generation $(\epsilon_n, \delta_n)$, #sections $m$, instruction $I$, noise variance $\sigma_{\text{emb}}$, #candidates $k$
**Output:** Synthetic note $X^{\text{syn}}$ with $(\epsilon, \delta)$-DP guarantee

1: $\text{SEC}^{\text{src}}_{1:m} \leftarrow \text{SECSPLIT}(X^{\text{src}})$                **// 3.2 Format and Term Identification**
2: $T^{\text{src}}_{1:m} \leftarrow \text{TERMEXT}(\text{SEC}^{\text{src}}_{1:m})$
3: $E^{\text{src}}_{1:m}, E_{\text{public}} \leftarrow \text{EMB}(T^{\text{src}}_{1:m}), \text{EMB}(D_{\text{public}})$         **// 3.3 Clinical Terms Generation**
4: **if** train **then**
5:     $\theta_t, \theta_p \leftarrow \mathcal{L}(\text{TERMGEN}_{\theta_t}(\text{PROJ}_{\theta_p}(E_{\text{public}} + \mathcal{N}(0, \sigma_{\text{emb}}))), D_{\text{public}})$       **// Train on $D_{\text{public}}$**
6: **else**
7:     $T^{\text{syn}}_{1:m} \sim \text{TERMGEN}_{\theta_t}(\text{PROJ}_{\theta_p}(\text{DPRP}^*(E^{\text{src}}_{1:m}, \frac{\epsilon_t}{m}, \frac{\delta_t}{m})))$        **// Infer for $X^{\text{src}}$**
8: **end if**
9: **if** train **then**
10:     $\theta_n \leftarrow \text{FastDP}(\mathcal{L}(\text{NOTEGEN}_{\theta_n}(I, \text{SEC}^{\text{src}}_{<i}, T^{\text{src}}_i), X^{\text{src}}), \epsilon_n, \delta_n)$   **// 3.4 Clinical Note Generation**
11: **end if**
12: $\text{SEC}^{\text{syn}}_j[i] \sim \text{NOTEGEN}_{\theta_n}(I, \text{SEC}^{\text{syn}}_{<j}[i], T^{\text{src / syn}}_j)$ **for** $i = 1, \ldots, k; j = 1, \ldots, m$
13: $X^{\text{syn}} \leftarrow \arg\min_{i \in [k]} \text{PPL}(\text{SEC}^{\text{syn}}_{1:m}[i])$            **// 3.5 DP Quality Maximiser**
14: **return** $X^{\text{syn}}$

---

TERMEXT, to each section to identify clinically salient terms, resulting in a list of section-specific medical terms: $[T^{\text{src}}_1, ..., T^{\text{src}}_m] = [\text{TERMEXT}(\text{SEC}^{\text{src}}_1), ..., \text{TERMEXT}(\text{SEC}^{\text{src}}_m)]$.

### 3.3 CLINICAL TERMS GENERATION

Clinical terms from private notes may still contain sensitive information. For example, unique combinations of diagnoses and procedures could re-identify patients. To mitigate this, we introduce an optional DP step for term generation. We formulate it as a reconstruction task, fine-tuning a generative model on the public term dataset $D_{\text{public}}$ to recover term lists from embeddings, and then applying it to private data under DP constraints. Specifically, any given (section-wise) term list is first embedded using a clinical text embedder, denoted EMB. A projection layer PROJ, parameterised by $\theta_p$, maps the embeddings to the hidden dimensionality required by the generative model TERMGEN, parameterised by $\theta_t$. The model then reconstructs the original term list from the projected embeddings.

To protect privacy when applying on private data, we adapt the DPRP schema (Gondara & Wang, 2020), a model-agnostic DP mechanism originally proposed for tabular data, to term embeddings derived from private notes, denoted as DPRP$^*$. The procedure perturbs private embeddings in four steps: (1) add dimension-wise random noise to input embeddings; (2) compute the covariance matrix of the input embeddings and add random noise; (3) perform singular value decomposition (SVD) on the noisy covariance matrix; (4) reconstruct the inputs from the noisy embeddings and the right singular vectors. The pseudocode of DPRP$^*$ is provided in Appendix B. To minimise the distribution difference between embeddings used in training and those perturbed during inference, we additionally add Gaussian noise to the embeddings during training. Formally, the process is defined as follows:

$$E' = \begin{cases} \text{EMB}(D_{\text{public}}) + \mathcal{N}(0, \sigma_{\text{emb}}), & \text{if training,} \\ \text{DPRP}^*(\text{EMB}(T^{\text{src}}), \frac{\epsilon_t}{m}, \frac{\sigma_t}{m}), & \text{otherwise.} \end{cases} \tag{1}$$

$$T^{\text{syn}} \sim \text{TERMGEN}_{\theta_t}(\text{PROJ}_{\theta_p}(E')) \tag{2}$$

Here, $\epsilon_t$ and $\sigma_t$ denote the privacy parameters of DPRP$^*$, and $m$ is the number of sections in a single note. Since $E'$ is computed at the section level, the overall privacy cost for an entire note accumulates across sections. To account for this, we distribute the privacy budget evenly by scaling the cost for each section to $\frac{1}{m}$ of the total budget.

### 3.4 CLINICAL NOTE GENERATION

We define section-wise clinical note generation as a conditional text generation task. Given the task instruction $I$, the content of the previous (generated) sections $[\text{SEC}_1, ..., \text{SEC}_{i-1}]$, and a list

of clinical terms $T_i$ for the current section, a generative model NOTEGEN, parameterised by $\theta_n$, is trained to produce the $i$-th section of the note under $(\epsilon_n, \delta_n)$-DP. During inference, each section is sampled sequentially as:

$$\text{SEC}_i^{\text{syc}} \sim \text{NOTEGEN}_{\theta_n}(I, [\text{SEC}_1^{\text{syn}}, ..., \text{SEC}_{i-1}^{\text{syn}}], T_i) \tag{3}$$

Here, $T_i$ can be the original extracted terms $T_i^{\text{src}}$ or the synthetic privatised terms $T_i^{\text{syn}}$, depending on the privacy configuration. With the section group named provided, the instruction $I$ is defined as shown in Figure 1. Finally, a synthetic full note is obtained by concatenating the generated sections: $X^{\text{syn}} = [\text{SEC}_1^{\text{syn}}, ..., \text{SEC}_m^{\text{syn}}]$.

## 3.5 DP QUALITY MAXIMISER

To improve the quality of the synthetic data, we introduce a quality maximisation strategy during inference by leveraging the generative capabilities of LLMs. Specifically, instead of generating a single synthetic note, we perform preference sampling on $k$ candidate notes for $X^{\text{src}}$, denoted $X^{\text{syn}}[1:k]$. Notably, this sampling procedure preserves the DP guarantee due to the post-processing property of DP. To select the most fluent and coherent output among the candidates, we use perplexity as the preference model. Perplexity reflects the likelihood of a sequence under an LLM, computed as the exponentiated average negative log-likelihood of the tokens. Lower perplexity indicates higher linguistic plausibility. To avoid bias from the generator itself, we compute perplexity scores using a reference domain-specialised LLM, denoted $\text{LLM}_{\text{ppl}}$. This ensures a more objective assessment of sequence quality. Formally, the perplexity of a candidate note $X^{\text{syn}}[i]$ is given by $\text{PPL}(X^{\text{syn}}[i]) = \exp\left(-\frac{1}{t}\sum_{i=1}^{t} log\, \text{LLM}_{\text{ppl}}(d_i|d_{<i})\right)$, where $d_{1,...,t}$ are tokens in $X^{\text{syn}}[i]$. The synthetic note with the lowest perplexity score is selected.

## 3.6 PRIVACY ANALYSIS

The overall privacy guarantee of Term2Note depends on the composition of its two DP components: TERMGEN and NOTEGEN. Specifically, TERMGEN is trained on $D_{\text{public}}$ and can optionally be applied to privatise terms for $D_{\text{test}}^{\text{src}}$, while NOTEGEN is trained on $D_{\text{train}}^{\text{src}}$ under DP. Since TERMGEN and NOTEGEN operate on disjoint subsets of private data, the overall privacy loss can be computed using parallel composition. Formally, the total privacy guarantee $(\epsilon, \delta)$ is defined as below, and the proof is provided in Appendix C.

$$(\epsilon, \delta) = \begin{cases} (\epsilon_n, \delta_n), & \text{if } T_i = T_i^{\text{src}}, \\ (\max(\epsilon_n, \epsilon_t), \max(\delta_n, \delta_t)), & \text{if } T_i = T_i^{\text{syn}}. \end{cases} \tag{4}$$

## 3.7 IMPLEMENTATION DETAILS

**SECSPLIT** To segment clinical notes into meaningful sections, we begin by considering the formatting conventions commonly found in clinical documentation. Although the SOAP format is widely adopted, it often requires manual annotation for accurate segmentation (Gao et al., 2022), limiting its applicability in automated processing. Moreover, there is no universally standardised format applicable across healthcare systems or institutions globally. To address this, we perform a preliminary analysis of the original clinical notes and develop a rule-based segmentation strategy using regular expression (regex) to automatically identify section titles. The span of each section is determined greedily, based on the position of a detected title and the nearest subsequent section title. A list of commonly occurring section titles is automatically curated, and we further group them into six broader semantic categories: *"Patient Information", "Clinical Course & History", "Examinations & Findings", "Laboratory & Imaging Results", "Hospital Stay & Treatment"*, and *"Medications & Discharge Plan"*. This taxonomy forms the basis for SECSPLIT, which splits each clinical note into at most six standardised sections corresponding to these categories. The complete list of extracted section titles and their groupings is provided in Appendix D.

**TERMEXT** Various biomedical terminology vocabularies exist, depending on the taxonomy adopted. In this work, we focus exclusively on terms from SNOMED CT, a comprehensive clinical vocabulary widely used in electronic health records (EHRs). Notably, SNOMED CT is also

included within the Unified Medical Language System (UMLS) (U.S. NLM, 2025), a metathesaurus that integrates multiple biomedical vocabularies. To extract medical terms from clinical text, we use QuickUMLS (Soldaini, 2016), an unsupervised tool for fast, approximate string matching against UMLS concepts. Following extraction, we retain only the SNOMED CT concepts.

**Backbone Models** For clinical term embedding, we use MedEmbed-large (Balachandran, 2024) as our embedder **EMB**. This encoder-only model is specifically fine-tuned for medical and clinical texts, making it well-suited for embedding domain-specific terms. For the two generative modules in Term2Note, we adopt lightweight yet effective language models: GPT2-Large (Radford et al., 2019) for the term generation model **TERMGEN**, and Llama-3.2-1B (Meta, 2024) or Gemma-3-1B (Google, 2025) for the note generation model **NOTEGEN**. Both Llama and Gemma are the most widely adpoted open-source LLMs, enabling reproducible evaluation under well-supported architectures. Furthermore, to increase architectural diversity and reduce the risk of model-specific bias, we include these two distinct model families. To compute the perplexity of generated notes, we use Asclepius-Llama3-8B (Kweon et al., 2024b) as our reference model **LLM$_{ppl}$**. This model is also pre-trained on clinical text, mitigating domain mismatch and providing reliable fluency estimates. Additionally, it supports a maximum input length of 8192 tokens, which is sufficient to accommodate the full length of most generated clinical notes.

# 4 EXPERIMENTAL SETTING

**Datasets** The MIMIC dataset series is one of the most widely used resources for clinical NLP. In this work, we use discharge notes from two MIMIC datasets for different purposes. MIMIC-III (Johnson et al., 2016) is used as the *public* dataset to train the term generation model **TERMGEN**, and MIMIC-IV notes (Johnson et al., 2023) serve as our *private* dataset for training the note generation model **NOTEGEN** under DP constraints. For both datasets, we apply a filtering step to exclude discharge notes that do not have any associated ICD codes. In addition, we exclude all notes annotated in the SNOMED CT Entity Linking Challenge (Hardman et al., 2025) from MIMIC-IV notes, as we reserve this subset as our test set. As a result, we construct the following three datasets for our experiments and summarises their statistics in Appendix E: $D_{\text{public}}$ contains around 52.7k notes derived from MIMIC-III where 500 notes are held out to assist with model development and validation; $D_{\text{train}}^{\text{src}}$ consists of around 122k notes derived from MIMIC-IV (excluding SNOMED notes); $D_{\text{test}}^{\text{src}}$ composes of 204 SNOMED notes.

**Hyperparameters** Following common practice in previous work (Yu et al., 2021; De et al., 2022; Baumel et al., 2024), we experiment with different privacy budgets by varying the overall $\epsilon \in [2, 5, 8]$. Accordingly, the privacy budget for term generation ($\epsilon_t$) or note generation ($\epsilon_n$) is set to one of these values. Following previous work, the corresponding $\delta$ value is set as $\frac{1}{N \log N}$ where $N$ denotes the size of the private dataset. All experiments are conducted on up to two Nvidia A100 80GB GPUs. More training details, including learning rate, number of epochs, batch size, etc., are provided in Appendix F.

**Baselines** We compare our proposed method against existing DP approaches for synthetic text generation. Specifically, we consider the following baselines: (1) DP-SGD with control codes (Yue et al., 2023): fine-tuning a language model under DP constraints using DP-SGD, where task-relevant control codes are prepended to the input. To ensure consistency with the DP training setup used in Term2Note, we adapt this method to use the FastDP algorithm. (2) AUG-PE (Xie et al., 2024): a recent method based on private evaluation (PE), designed to generate synthetic text without requiring model training. It leverages a pretrained LLM to produce an initial pool of candidate synthetic texts. These candidates are then evaluated under the PE mechanism, which privately measures their semantic proximity to the real private texts using a DP distance function. Only candidates that fall within a DP-valid similarity threshold are retained. The model is subsequently prompted to produce additional samples conditioned on these retained examples, iteratively enlarging the synthetic corpus while maintaining differential privacy. For a fair comparison with Term2Note, we adapt both baselines to the clinical note generation task. Specifically, we prepend ICD codes associated with each note as control codes to guide the generation process, aligning with the conditioning setup used in our framework.

| | | Fidelity | | | | Utility | | | | | |
|---|---|---|---|---|---|---|---|---|---|---|---|
| **Method** | | **Length** | **Unary/Binary Term** | | **Semantic** | | **F1** | | **AUC** | | **Precision@$k$** |
| | | **KL Div.↓** | **Jaccard↑** | **KL Div.↓** | **MAUVE↑** | | **Micro** | **Macro** | **Micro** | **Macro** | $k=3$ $k=5$ |
| Original Data | | | | | | | 57.03 | 30.80 | 82.01 | 58.88 | 68.93 62.14 |
| | | | | | $\epsilon = \infty$ | | | | | | |
| AUG-PE | | 11.96 | 0.14/0.02 | 7.59/16.34 | 0.01 | | 45.82 | 14.84 | 79.52 | 54.35 | 68.48 61.77 |
| FastDP | | 1.04 | **0.53/0.28** | 0.32/**0.84** | 0.12 | | 53.02 | 25.51 | 79.35 | 51.05 | 69.77 60.39 |
| Term2Note ($\epsilon_n = \infty$) | | 0.25 | 0.52/0.20 | **0.22**/1.08 | **0.59** | | 49.95 | 21.89 | **81.40** | **55.43** | 69.77 61.96 |
| ($\epsilon_n = \infty, \epsilon_t = \infty$) | | 0.68 | 0.43/0.14 | 0.45/1.95 | 0.46 | | 49.24 | 21.90 | 80.24 | 51.54 | 67.81 61.08 |
| ($\epsilon_n = 8, \epsilon_t = \infty$) | | 0.26 | 0.39/0.13 | 0.50/1.19 | 0.35 | | 48.16 | 20.63 | 78.72 | 50.01 | 65.35 58.72 |
| ($\epsilon_n = 5, \epsilon_t = \infty$) | | 0.28 | 0.38/0.13 | 0.53/1.23 | 0.27 | | 51.00 | 22.73 | 78.80 | 50.37 | 67.15 59.91 |
| ($\epsilon_n = 2, \epsilon_t = \infty$) | | 0.43 | 0.37/0.12 | 0.60/1.35 | 0.36 | | 48.57 | 20.31 | 79.56 | 51.75 | 68.64 60.41 |
| | | | | | $\epsilon = 8$ | | | | | | |
| AUG-PE | | 11.71 | 0.19/0.03 | 5.03/12.18 | 0.01 | | 40.73 | 13.28 | 78.13 | 53.49 | 63.24 59.03 |
| FastDP | | 4.51 | 0.31/0.10 | 2.88/5.88 | 0.02 | | 48.58 | 16.40 | 80.74 | 51.57 | 69.79 61.59 |
| Term2Note ($\epsilon_n = 8$) | | 0.39 | 0.40/0.13 | 0.47/1.14 | 0.53 | | 49.71 | 21.28 | 80.03 | 52.80 | 67.49 61.48 |
| ($\epsilon_n = 8, \epsilon_t = 8$) | | 0.16 | 0.38/0.13 | 0.62/1.15 | 0.37 | | 52.31 | 26.50 | 78.19 | 50.17 | 67.81 57.36 |
| ($\epsilon_n = 8, \epsilon_t = 5$) | | **0.15** | 0.39/0.13 | 0.64/1.16 | 0.46 | | 49.52 | 21.36 | 79.08 | 50.11 | 68.29 60.68 |
| ($\epsilon_n = 8, \epsilon_t = 2$) | | 0.19 | 0.38/0.12 | 0.61/1.18 | 0.38 | | 53.25 | 24.66 | 78.85 | 49.41 | 68.31 59.91 |
| | | | | | $\epsilon = 5$ | | | | | | |
| AUG-PE | | 11.70 | 0.11/0.01 | 7.75/13.58 | 0.01 | | 48.10 | 17.44 | 77.78 | 53.33 | 63.01 56.94 |
| FastDP | | 3.40 | 0.29/0.09 | 2.97/5.67 | 0.04 | | 49.30 | 16.23 | 80.54 | 54.22 | 67.31 **61.98** |
| Term2Note ($\epsilon_n = 5$) | | 0.20 | 0.41/0.14 | 0.42/1.17 | 0.39 | | 47.94 | 20.31 | 79.29 | 51.19 | 66.04 61.69 |
| ($\epsilon_n = 5, \epsilon_t = 5$) | | 0.20 | 0.38/0.13 | 0.63/1.19 | 0.43 | | **54.83** | **28.96** | 78.20 | 50.32 | 64.56 57.18 |
| ($\epsilon_n = 5, \epsilon_t = 2$) | | 0.36 | 0.38/0.12 | 0.64/1.32 | 0.36 | | 51.26 | 21.45 | 79.05 | 50.44 | 66.36 60.49 |
| | | | | | $\epsilon = 2$ | | | | | | |
| AUG-PE | | 12.11 | 0.19/0.03 | 5.42/11.85 | 0.01 | | 40.90 | 13.57 | 78.29 | 53.38 | 63.74 60.10 |
| FastDP | | 9.67 | 0.14/0.03 | 5.79/10.89 | 0.01 | | 51.06 | 20.04 | 79.98 | 51.31 | 66.32 59.99 |
| Term2Note ($\epsilon_n = 2$) | | 0.43 | 0.39/0.12 | 0.48/1.17 | 0.31 | | 51.78 | 23.36 | 79.00 | 50.60 | 67.00 59.52 |
| ($\epsilon_n = 2, \epsilon_t = 2$) | | 0.48 | 0.37/0.12 | 0.68/1.29 | 0.31 | | 51.87 | 23.06 | 79.43 | 51.30 | 69.45 60.31 |

*(Left margin label spanning the data rows: **Synthetic**)*

Table 1: Fidelity and utility evaluation of synthetic datasets generated by different methods on Llama-3.2-1B. Utility metrics are reported with the average score across 5-fold cross-validation. The **best result** among all methods to generate synthetic datasets is shown in bold, and the best result at the same privacy cost is underlined. Additional results on Gemma-3-1B, and evaluation results, including the standard deviations of utility metrics, precision and recall scores, are provided in Appendix I.

**Evaluation** The automatic evaluation of synthetic data typically encompasses three key aspects: fidelity, privacy, and utility. In this work, we mainly focus on fidelity and utility. While privacy is formally guaranteed through DP, we additionally conduct a preliminary empirical privacy analysis to assess potential leakage risks. The results of this experiment show encouraging outcomes and are presented in Appendix G.

**Fidelity** assesses how closely synthetic notes resemble real data across structural, syntactic, and semantic dimensions. Structural similarity is measured by comparing text length distributions using Kullback–Leibler (KL) divergence. Syntactic similarity focuses on clinical term overlap, evaluated via Jaccard similarity and KL divergence over unary and binary term sets, where binary terms refer to co-occurring pairs in the same note. Semantic similarity is evaluated using MAUVE (Pillutla et al., 2021). MAUVE assesses global distributional fidelity in a semantic embedding space, which we use as a proxy for high-level semantic similarity between synthetic and real notes. BioMistral-7B (Labrak et al., 2024) serves as the language model to ensure domain-relevant representations.

**Utility** evaluates the usefulness of the synthetic data for downstream clinical applications, where we assess the performance of models trained on synthetic notes and tested on real data. Specifically, we consider **ICD coding prediction** as the downstream task, leveraging the ICD labels available in the private dataset. The ICD coding task is defined as follows: given a discharge note, the model is required to predict all applicable ICD codes associated with it. Due to the large number of fine-grained codes in both ICD-9 and ICD-10, direct prediction is highly challenging. To simplify the task, we normalise all ICD codes to their chapter-level categories and further merge the ICD-9 and ICD-10 codes into a unified set of 20 code groups. The complete mapping from ICD-9/10 codes to these unified categories is provided in Appendix H. We assess the utility of synthetic notes by comparing model performance in two training settings: (1) Train-Real-Test-Real, where the model is both

trained and tested on the original dataset; (2) Train-Synthetic-Test-Real, where the model is trained on the synthetic dataset and tested on the original dataset. We report standard multi-label classification metrics, including the micro and macro average of F1 score and AUC, and Precision@$k$. To handle the long input sequences, we adopt Clinical-Longformer (Li et al., 2023b) as our classifier. This model supports input lengths exceeding 4k tokens and is pre-trained on clinical corpora, making it well-suited for our task. For evaluation, we use the private test set $D_{\text{test}}^{\text{src}}$, which contains 204 notes in total. We further split it into a training subset $D_{\text{test-train}}^{\text{src}}$ and a testing subset $D_{\text{test-test}}^{\text{src}}$. The classifier is trained on $D_{\text{test-train}}^{\text{src/syn}}$ and evaluated on $D_{\text{test-test}}^{\text{src}}$. Due to the limited dataset size, we employ 80:20 train-test 5-fold cross-validation.

**Human Evaluation** is additionally conducted with three licensed physicians to assess the clinical quality of the generated notes. The evaluation follows a pairwise comparison protocol, where each physician is presented with a randomly selected pair of notes and asked to indicate which one is clinically better. The pairs are sampled from outputs generated by three different models under $\epsilon = 8$, as well as from the original (real) notes. Each physician evaluates a minimum of 100 pairs, resulting in a total of 412 pairwise comparisons across all annotators. Based on these annotations, we estimate pairwise model preferences and infer a global ranking using the Bradley-Terry (BT) (Bradley & Terry, 1952) model. The BT model also allows us to estimate the probability that one model $\mathcal{M}_1$ is preferred over another model $\mathcal{M}_2$, based on the aggregated comparison outcomes.

## 5 RESULTS

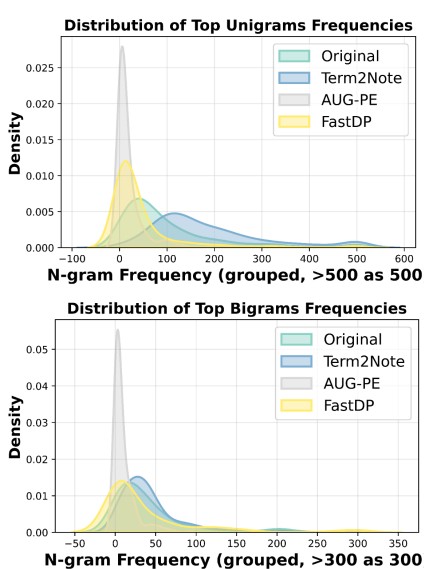

Figure 2: Distribution of n-gram frequencies in clinical notes generated by different DP methods under $\epsilon = 8$. Note: density estimates may extend below zero due to smoothing; all observed frequencies are positive integers.

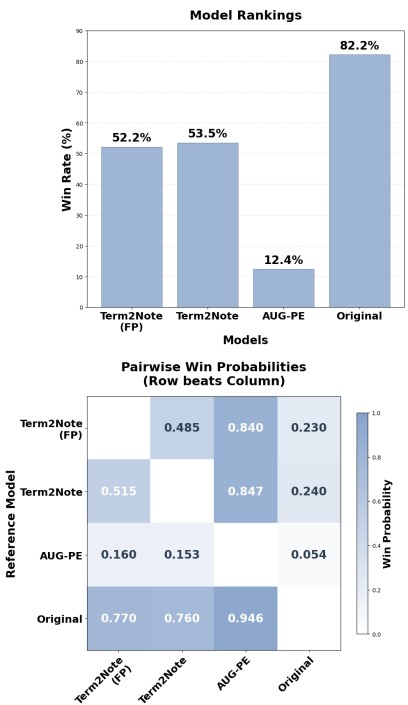

Figure 3: Human evaluation results summarised using the BT model. Term2Note (FP) denotes the *full privacy* setting, where both terms and notes are synthesised under DP constraints.

**Term2Note consistently achieves better structural, syntactic, and semantic similarity to the original data, despite operating under stronger privacy constraints and fewer assumptions.** In terms of *structural similarity*, Term2Note achieves the lowest KL divergence in text length distribution (as low as 0.15), indicating faithful preservation of note structure. For *syntactic similarity*, it obtains the (almost) highest Jaccard scores for both unary and binary clinical terms, alongside the lowest KL divergence in term frequency distribution, suggesting close alignment with real clinical

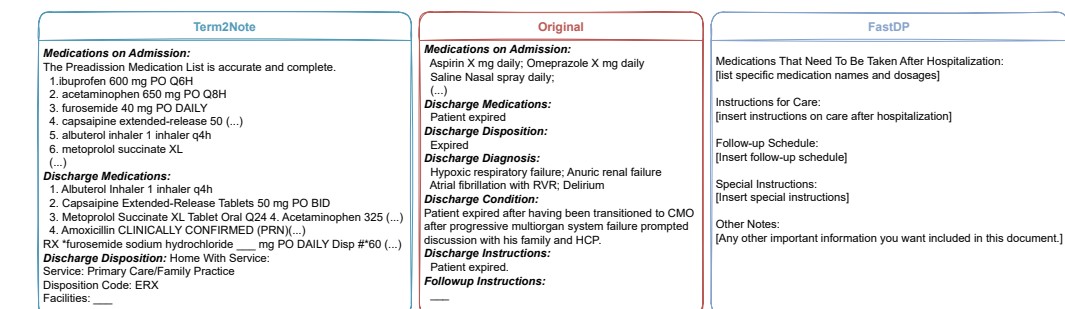

Figure 4: Examples of the last section in clinical notes generated by different models. For illustration purposes, some content is redacted with "(...)", and numeric values in the original note are de-identified. No other modifications were made.

content. In *semantic space*, Term2Note substantially outperforms both baseline methods in MAUVE score, confirming that its generated text is significantly more aligned with the original notes in the semantic space. These advantages remain even under a strict privacy budget of $\epsilon = 2$ and when further enforcing DP on clinical term generation (i.e., $\epsilon_t = 2$), underscoring the robustness of the approach. Additionally, Figure 2 shows the distribution of n-gram frequencies, where Term2Note exhibits a distribution more closely aligned with the original notes compared to the baseline methods. More results are presented in Appendix I.

**Regarding utility, Term2Note demonstrates strong overall performance and consistently preserves clinical utility across varying privacy budgets, even under stricter constraints on both term and note generation.** Across all privacy levels, Term2Note achieves the highest F1 scores, outperforming both AUG-PE and FastDP, and showing the closest performance to the original data. While AUC and Precision@k scores are comparable, rather than uniformly superior, to those of AUG-PE and FastDP, this variation reflects the different aspects of model behaviour captured by each metric. Notably, even when the privacy budget for note generation is reduced to $\epsilon_n = 2$ and additional constraints are applied to privatise clinical terms (with $\epsilon_t = 2$), Term2Note maintains strong utility, with F1 and AUC scores comparable to or better than the baselines operating under looser privacy conditions.

**Term2Note is consistently preferred by human experts over AUG-PE, with minimal quality loss under full privacy.** As shown in Figure 3, Term2Note achieves a win rate of 52.2%–53.5% across DP settings, substantially outperforming AUG-PE (12.4%). The pairwise win probabilities (lower panel) further demonstrate that Term2Note reliably outperforms AUG-PE across all conditions. Importantly, introducing full privacy constraints, where both clinical terms and notes are protected, has only a marginal effect on human preference. These results suggest that Term2Note maintains high perceived quality while offering stronger privacy guarantees.

**Term privatisation maintains a good level of semantic coherence while effectively abstracting away from potentially privacy-leaking details.** We evaluate the semantic alignment of the term generation model **TERMGEN** on $D_{\text{test}}^{\text{src}}$ by computing the cosine similarity between the embeddings of original and generated term lists, using the clinical term encoder **EMB**. Without the DP mechanism DPRP*, the mean cosine similarity is high (0.82), indicating strong recovery of original terms. When DPRP* is applied, the mean similarity drops to 0.61, reflecting the expected privacy-induced noise. This drop indicates that the generated terms stay semantically coherent without closely matching the originals, reducing the risk of revealing sensitive information (see Appendix J for case studies).

**Qualitative Analysis** To further assess the quality of synthetic notes, two physicians each reviewed 20 samples generated by Term2Note, covering both standard and full privacy settings with $\epsilon = 5$. As intended by the design of the evaluation, where physicians were explicitly asked to identify any potential clinical issues, the feedback focuses on shortcomings rather than general plausibility. Importantly, not all notes contained identifiable issues. In one physician's review, 45% were

judged to have no clinical problems, indicating that a substantial proportion of generations were considered clinically plausible. Both physicians agreed that many synthetic notes were plausible as discharge summaries and generally exhibited sound structural organisation. However, recurring issues emerged around clinical accuracy and coherence. The first physician highlighted problems such as missing or misordered sections, internal inconsistencies (e.g., conflicting medications), and vague or overly generic phrasing; under full privacy constraints, repetition was more common. The second physician, who reviewed a different set of examples, reported more content-level issues, including medication misclassifications (e.g., labelling omeprazole as an antibiotic), illogical or irrelevant narrative insertions, and errors in clinical reasoning. These observations suggest that while Term2Note performs well in preserving structural fidelity, improvements are needed in clinical fact consistency and terminology use. Figure 4 illustrates examples of final note sections, demonstrating the model's ability to maintain coherence in longer contexts. Although Term2Note occasionally omits sections, its outputs more closely align with the structure of the original note compared to the baseline model.

The qualitative feedback highlights important areas where clinical reliability can be further strengthened. A key challenge lies in ensuring factual accuracy and preventing clinical inconsistencies (e.g., incorrect medication classes or contradictory clinical states). One promising direction is to integrate clinical fact-checking mechanisms into the generation process. Another complementary approach is to incorporate structured consistency checks, such as verifying that medications align with listed conditions, either as part of the decoding process or through post-generation filtering. Finally, incorporating lightweight clinical reasoning or rule-based validators could help detect illogical narrative transitions and prevent contractions across sections. We view these techniques as natural extension of Term2Note and plan to explore them in future work to further enhance clinical usefulness and safety.

## 6 CONCLUSIONS

In this paper, we introduce Term2Note, a novel framework for DP clinical note generation by synthesising section-wise clinical content conditioned on medical terms while providing formal privacy guarantees. Experimental results demonstrate that Term2Note consistently outperforms existing baselines by a substantial margin. It achieves the highest fidelity, closely matching original notes in terms of structure, semantics, and medical term distribution. Furthermore, Term2Note attains comparable utility to real notes on a downstream ICD coding task, confirming the practical effectiveness of the synthetic data. Human evaluation further supports the superiority of Term2Note, showing that clinical experts consistently prefer its outputs over those of baseline models. Overall, Term2Note provides a promising and principled solution to the data scarcity problem in healthcare NLP, enabling generating of high-quality, privacy-preserving synthetic clinical notes, facilitating privacy-conscious data sharing.

## REPRODUCIBILITY STATEMENT

To ensure reproducibility, we have taken several steps across the main text, appendix, and supplementary materials. First, we provide pseudocode for Term2Note in Algorithm 1, which outlines the core components of our method in a concise and implementation-ready format. Second, Section 4 offers a detailed description of the dataset preprocessing pipeline, including normalisation procedures, filtering criteria, and the grouping of sections (with additional details in Appendix D). This section (along with Appendix F) also specifies all hyperparameters used in training and evaluation, as well as the computational resources required to reproduce our experiments. Third, we report the evaluation protocol in detail, including the choice of evaluation model, metrics, and sampling strategies, to make our experimental setup transparent. Fourth, Appendix C provides a complete proof of our privacy guarantees, with all assumptions and derivations made explicit. Finally, we include the full source code and experiment scripts as supplementary materials to facilitate direct replication and extension of our results.

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

## A   THE USE OF LARGE LANGUAGE MODELS (LLMs)

LLMs were used solely as a general-purpose writing aid. The initial drafts were written by the authors, and LLMs were employed to polish grammar and improve coherence. All suggested edits were manually reviewed and selectively incorporated by the authors. LLMs did not contribute to research ideation, experimental design, implementation, or original writing beyond this assistive role.

## B   DPRP* ALGORITHM

Algorithm 2 presents the pseudocode for DPRP*.

---
**Algorithm 2** DPRP*

---
**Input:** Embeddings $E$, privacy parameters $(\epsilon, \delta)$ , privacy allocation $b = 0.85$
**Output:** Privatised Embeddings $E_{\text{DP}}$ with $(\epsilon, \delta)$-DP

1: $(\epsilon_1, \delta_1), (\epsilon_2, \delta_2) \leftarrow 0.85 * (\epsilon, \delta), 0.15 * (\epsilon, \delta)$
2: Derive $\sigma_i$ from $(\epsilon_{1i}, \delta_{1i}); i \in [1, 2]$
3: $E' = E + \mathcal{N}(0, \sigma_1^2)$
4: $E'_C = E^T E + \mathcal{N}(0, \sigma_2^2)$
5: $V' \Sigma' V'^T = \text{SVD}(E'_C)$
6: $V'_k = V'[1, ..., k]; k = 0.6 * E_{\text{hdim}}$
7: $E_{\text{DP}} = E' V'^{+}_k V'^T_k$              // + refers to the Moore-Penrose pseudoinverse
8: **return** $E_{\text{DP}}$

---

## C   PRIVACY PROOF

Recall our privacy analysis,

$$(\epsilon, \delta) = \begin{cases} (\epsilon_n, \delta_n), & \text{if } T_i = T_i^{\text{src}}, \\ (\max(\epsilon_n, \epsilon_t), \max(\delta_n, \delta_t)), & \text{if } T_i = T_i^{\text{syn}}. \end{cases}$$

When $T_i = T_i^{\text{src}}$, there is only one DP component, i.e., NOTEGEN which satisfies $(\epsilon_n, \delta_n)$-DP, therefore, the $(\epsilon, \delta) = (\epsilon_n, \delta_n)$, i.e., Term2Note satisfies $(\epsilon_n, \delta_n)$-DP.

*Proof.* We prove that for the full privatisation setting $(T_i = T_i^{\text{syn}})$, Term2Note achieves $(\max(\epsilon_n, \epsilon_t), \max(\delta_n, \delta_t))$-DP by applying the parallel composition theorem.

**Step 1: Parallel Composition Lemma**

First, we establish the parallel composition property.

**Lemma 3** (Parallel Composition)**.** Let dataset $D = D_1 \cup D_2$, and $D_1 \cap D_2 = \emptyset$. Let $\mathcal{M}_1 : \mathbb{N}^{|\mathcal{X}_1|} \to R_1$ be $(\epsilon_1, \delta_1)$-DP and $\mathcal{M}_2 : \mathbb{N}^{|\mathcal{X}_2|} \to R_2$ be $(\epsilon_2, \delta_2)$-DP. Then $\mathcal{M}(D) = (\mathcal{M}_1(D_1), \mathcal{M}_2(D_2))$ is $(\max(\epsilon_1, \epsilon_2), \max(\delta_1, \delta_2))$-DP.

*Proof of Lemma.* Let $D$ and $D'$ be neighboring datasets differing by one record. Since $D_1 \cap D_2 = \emptyset$, the differing record is in either $D_1$ or $D_2$, but not both.

**Case 1:** The differing record is in $D_1$, so $D_1 \neq D'_1$ but $D_2 = D'_2$.

For any measurable sets $B_1 \subseteq R_1, B_2 \subseteq R_2$:

$$
\begin{aligned}
&P[\mathcal{M}(D) \in B_1 \times B_2] \\
&= P[\mathcal{M}_1(D_1) \in B_1] \cdot P[\mathcal{M}_2(D_2) \in B_2] \\
&\leq (e^{\epsilon_1} P[\mathcal{M}_1(D_1') \in B_1] + \delta_1) \cdot P[\mathcal{M}_2(D_2) \in B_2] \\
&= (e^{\epsilon_1} P[\mathcal{M}_1(D_1') \in B_1] + \delta_1) \cdot P[\mathcal{M}_2(D_2') \in B_2] \\
&= e^{\epsilon_1} P[\mathcal{M}(D') \in B_1 \times B_2] + \delta_1 P[\mathcal{M}_2(D_2') \in B_2] \\
&\leq e^{\epsilon_1} P[\mathcal{M}(D') \in B_1 \times B_2] + \delta_1
\end{aligned}
$$

**Case 2:** The differing record is in $D_2$, so $D_1 = D_1'$ but $D_2 \neq D_2'$. Similarly:

$$
P[\mathcal{M}(D) \in B_1 \times B_2] \leq e^{\epsilon_2} P[\mathcal{M}(D') \in B_1 \times B_2] + \delta_2
$$

**Combining cases:** For arbitrary neighbouring datasets, we have:

$$
\begin{aligned}
&P[\mathcal{M}(D) \in B_1 \times B_2] \\
&\leq e^{\max(\epsilon_1, \epsilon_2)} P[\mathcal{M}(D') \in B_1 \times B_2] + \max(\delta_1, \delta_2)
\end{aligned}
$$

Therefore, $\mathcal{M}$ is $(\max(\epsilon_1, \epsilon_2), \max(\delta_1, \delta_2))$-DP. $\qquad\square$

**Step 2: Application to Term2Note**

Now we apply the parallel composition lemma to Term2Note.

We have:

- $\mathcal{M}_1 = \text{NOTEGEN}$ training on $D_{\text{train}}$, which is $(\epsilon_n, \delta_n)$-DP

- $\mathcal{M}_2 = \text{TERMGEN}$ processing on $D_{\text{test}}$, which is $(\epsilon_t, \delta_t)$-DP

- $D_{\text{train}} \cap D_{\text{test}} = \emptyset$

Term2Note can be written as:

$$
\text{Term2Note}(D) = f(\mathcal{M}_1(D_{\text{train}}), \mathcal{M}_2(D_{\text{test}}))
$$

where $f$ is a deterministic function that applies the trained NOTEGEN model to the synthetic terms from TERMGEN.

Since $f$ is a post-processing function applied to the outputs of the parallel composition, and post-processing preserves differential privacy, we have:

$$
\text{Term2Note}(D) \text{ is } (\max(\epsilon_n, \epsilon_t), \max(\delta_n, \delta_t))\text{-DP}
$$

$\qquad\square$

## D  SECTION GROUPING

Table 2 presents the section grouping taxonomy for our SECSPLIT.

## E  DATASET STATISTICS

Table 3 summarises key statistics of the three datasets used in this study, including the number of clinical notes, and average note length, among other relevant attributes.

| Group Name | Sections |
|---|---|
| Patient Information | "Name", "Unit No", "Admission Date", "Discharge Date", "Date of Birth", "Sex", "Service", "Allergies", "Attending" |
| Clinical Course & History | "Chief Complaint", "Major Surgical or Invasive Procedure", "History of Present Illness", "Review of Systems", "Past Medical History", "Social History", "Family History" |
| Examinations & Findings | "Physical Exam" |
| Laboratory & Imaging Results | "Pertinent Results" |
| Hospital Stay & Treatment | "Brief Hospital Course" |
| Medications & Discharge Plan | "Medications on Admission", "Discharge Medications", "Discharge Disposition, "Discharge Diagnosis", "Discharge Condition", "Discharge Instructions", "Followup Instructions" |

Table 2: The grouped section titles.

| Corpus | MIMIC-III | MIMIC-IV | |
|---|---|---|---|
| Dataset | $D_{\text{public}}$ | $D_{\text{train}}^{\text{src}}$ | $D_{\text{test}}^{\text{src}}$ |
| # notes | 52,722 | 122,202 | 204 |
| avg. # tokens | 3327.93 | 3360.60 | 2818.63 |
| avg. # sections | 4.59 | 5.77 | 5.79 |
| avg. # terms | 176.26 | 203.20 | 173.33 |
| avg. # ICD codes | - | - | 6.73 |

Table 3: Dataset statistics. avg. refers to the average of. # tokens is calculated by taking the average of tokens in each note, tokenised by Llama-3.2-1B-Instruct.

## F HYPERPARAMETERS

**TERMGEN** We fine-tune GPT-2-large on section-wise clinical terms extracted from $D_{\text{public}}$ for up to 5 epochs. The final model is selected based on the checkpoint with the highest F1 score, evaluated on a held-out set of 500 notes. During training, we set the embedding perturbation scale $\sigma_{\text{emb}} = 0.05$, with a batch size of 8 and a learning rate of 2e-5. At inference, we use a batch size of 16 and a maximum generation length of 512 tokens. To ensure reproducibility, decoding is performed with a temperature of 0.1 and top-$p$ set to 1.0.

**NOTEGEN** **Training:** We fine-tune Llama-3.2-1B-Instruct or Gemma-3-1B-IT on $D_{\text{train}}^{\text{src}}$ for up to 2 epochs using 2 GPUs. The batch size per device is 2, with a gradient accumulation step of 64 and a learning rate of 5e-5. We enable DeepSpeed ZeRO Stage 3 to optimise memory usage. **Inference:** We adapt vLLM for faster generation, with decoding parameters set as temperature = 0.1, top-$p$ = 1.0, repetition penalty = 1.2 and max tokens per section = 2048 across all experiments. Llama-3.2 tends to generate overly long outputs during section-wise generation, so we apply a logit bias on the EOS token to encourage early stopping. This bias is set between 0.5 and 6.0: for DP-enabled models, the value is 0.5 or 1.0; for the non-private setting ($\epsilon = \infty$), it is set to 6.0. Additionally, DP-enabled models use a frequency penalty of 0.4 to further discourage repetition.

For the FastDP baseline, which produces relatively short outputs, we apply only a repetition penalty during inference. Before applying the DP quality maximiser, we generate multiple candidates per input: 4 for Term2Note and FastDP, and 7 for AUG-PE, using the same decoding settings described above.

# G  DISTANCE-BASED PRIVACY EVALUATION

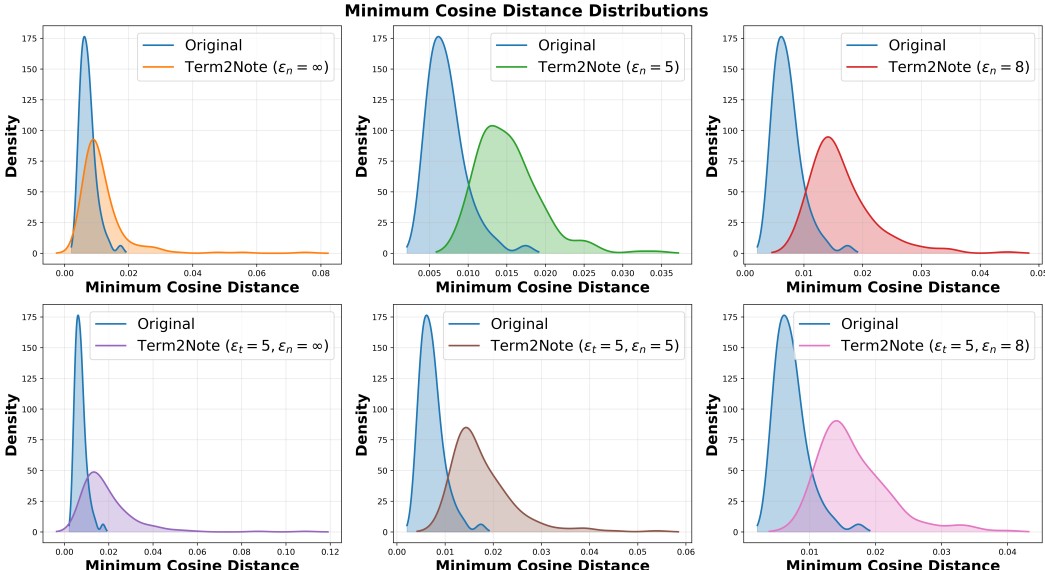

Figure 5: Distribution of minimum cosine distances for all evaluated synthetic strategies compared to the baseline of original test notes. Here, $\epsilon_t$ and $\epsilon_n$ are the privacy budgets for TERMGEN and NOTEGEN, respectively.

A preliminary privacy evaluation is conducted to assess the privacy-preserving properties of the synthetic clinical notes using a membership inference attack (MIA) framework. We compare the distribution of minimum cosine distances between the synthetic notes generated by Term2Note and the original training data to a baseline of real, non-member test notes. As illustrated in Figure 5, the synthetic notes are, on average, located significantly further from the training data than the test notes. Notably, the setting with $\epsilon_n = \infty$, which is generated without DP, exhibited the most overlap with the test set's distribution. This expected outcome highlights the privacy benefits of the other DP settings, and provides a clear baseline for comparison.

In future work, we intend to expand our privacy evaluation using canary-based membership inference attacks. This approach involves injecting specially crafted canaries into the training data to establish a worst-case lower bound on privacy risks.

# H  ICD CODES GROUPING

Table 4 shows the mapping between our combined ICD categories and the corresponding ICD-9 and ICD-10 chapter headings. For fine-tuning Clinical-Longformer on this classification task, we train for 30 epochs per setting (i.e., model and data fold), with a batch size of 8 and a learning rate of 2e-5.

# I  SUPPLEMENTARY EXPERIMENTAL RESULTS

## I.1  FIDELITY

Table 5 presents supplementary fidelity evaluation results, including an ablation analysis of the DP Quality Maximiser. These results further support the effectiveness of our approach in preserving structural and semantic fidelity under DP constraints.

**Larger Model**  We present preliminary results using a larger model, Llama-3.3-70B, evaluated with two non-private methods: Retrieval-Augmented Generation (RAG) and LoRA-based fine-tuning. In the RAG setup, we retrieve the top-5 most similar sections from the training set $D_{\text{train}}^{\text{src}}$

| Combined ICD Category | ICD-9 | ICD-10 |
|---|---|---|
| Certain Infectious And Parasitic Diseases | Infectious And Parasitic Diseases | Certain Infectious And Parasitic Diseases |
| Neoplasms | Neoplasms | Neoplasms |
| Endocrine, Nutritional And Metabolic Diseases, And Immunity Disorders | Endocrine, Nutritional And Metabolic Diseases, And Immunity Disorders | Endocrine, Nutritional And Metabolic Diseases |
| Diseases Of The Blood And Blood-Forming Organs And Certain Disorders Involving The Immune Mechanism | Diseases Of The Blood And Blood-Forming Organs | Diseases Of The Blood And Blood-Forming Organs And Certain Disorders Involving The Immune Mechanism |
| Mental And Behavioural Disorders | Mental Disorders | Mental And Behavioural Disorders |
| Diseases Of The Nervous System And Sense Organs | Diseases Of The Nervous System And Sense Organs | Diseases Of The Nervous System |
| Diseases Of The Circulatory System | Diseases Of The Circulatory System | Diseases Of The Circulatory System |
| Diseases Of The Respiratory System | Diseases Of The Respiratory System | Diseases Of The Respiratory System |
| Diseases Of The Digestive System | Diseases Of The Digestive System | Diseases Of The Digestive System |
| Diseases Of The Genitourinary System | Diseases Of The Genitourinary System | Diseases Of The Genitourinary System |
| Complications Of Pregnancy, Childbirth, And The Puerperium | Complications Of Pregnancy, Childbirth, And The Puerperium | Complications Of Pregnancy, Childbirth, And The Puerperium |
| Diseases Of The Skin And Subcutaneous Tissue | Diseases Of The Skin And Subcutaneous Tissue | Diseases Of The Skin And Subcutaneous Tissue |
| Diseases Of The Musculoskeletal System And Connective Tissue | Diseases Of The Musculoskeletal System And Connective Tissue | Diseases Of The Musculoskeletal System And Connective Tissue |
| Congenital Malformations, Deformations And Chromosomal Abnormalities | Congenital Anomalies | Diseases Of The Musculoskeletal System And Connective Tissue |
| Congenital Malformations, Deformations And Chromosomal Abnormalities | - | Congenital Malformations, Deformations And Chromosomal Abnormalities |
| Certain Conditions Originating In The Perinatal Period | Certain Conditions Originating In The Perinatal Period | Certain Conditions Originating In The Perinatal Period |
| Symptoms, Signs And Abnormal Clinical And Laboratory Findings, Not Elsewhere Classified | Symptoms, Signs, And Ill-Defined Conditions | Symptoms, Signs And Abnormal Clinical And Laboratory Findings, Not Elsewhere Classified |
| Injury, Poisoning And Certain Other Consequences Of External Causes | Injury And Poisoning | Injury, Poisoning And Certain Other Consequences Of External Causes |
| External Causes Of Morbidity And Mortality, Injusy and Poisoning | External Causes Of Injury And Poisoning | External Causes Of Morbidity And Mortality, Injusy and Poisoning |
| Factors Influencing Health Status And Contact With Health Services | Factors Influencing Health Status And Contact With Health Services | Factors Influencing Health Status And Contact With Health Services |
| Diseases Of The Eye And Adnexa | - | Diseases Of The Eye And Adnexa |
| Diseases Of The Ear And Mastoid Process | - | Diseases Of The Ear And Mastoid Process |
| Codes For Special Purposes | - | Codes For Special Purposes |

Table 4: The grouped ICD codes.

to assist section-wise generation. While neither approach offers privacy guarantees, they serve as reference points for performance with large-scale models. As shown in the results, fine-tuning significantly outperforms RAG, highlighting the importance of parameter adaptation for note synthesis. However, the high computational cost of fine-tuning such large models motivates our focus on efficient methods based on smaller models, such as the 1B-parameter version used in Term2Note.

**DP Quality Maximiser**   We evaluate the effectiveness of our proposed DP quality maximiser on models trained with $\epsilon = 8$. As shown in the results, it consistently improves MAUVE scores for both Term2Note and FastDP. For FastDP, improvements extend across all fidelity metrics, highlighting the value of the maximiser in enhancing output quality under DP constraints.

Beyond perplexity, we investigate a range of reference-free (RF) metrics to guide the selection of high-quality generations, including maximum and mean sentence length (in words and characters), self-BLEU, and distinct-$n$ variants. To evaluate these metrics, we manually annotate a small set of synthetic sections as "good" or "bad" based on readability, with approximately 12% labelled as "bad". Metrics are evaluated on their ability to identify these poor-quality sections via scalar thresh-

| | Length | | Unary/Binary Term | | Semantic |
|---|---|---|---|---|---|
| | **mean** | **KL Div.**↓ | **Jaccard**↑ | **KL Div.**↓ | **MAUVE**↑ |
| Original Data | 2819 | - | - | - | - |
| $\epsilon = \infty$ | | | | | |
| AUG-PE | 282 | 11.96 | 0.14/0.02 | 7.59/16.34 | 0.01 |
| Term2Note | 3552 | **0.25** | 0.52/0.20 | 0.22/1.08 | **0.59** |
| Term2Note (Llama-3.3-70b 4-bit) | 4115 | 0.87 | 0.55/0.23 | 0.17/2.23 | 0.38 |
| RAG (Llama-3.3-70b) | 3220 | 0.75 | 0.43/0.17 | 0.62/1.60 | 0.22 |
| $\epsilon = 8$ | | | | | |
| AUG-PE | 203 | 11.71 | 0.19/0.03 | 5.03/12.18 | 0.01 |
| FastDP | 961 | 4.51 | 0.31/0.10 | 2.88/5.88 | 0.02 |
| | 449.25±148.65 | 7.79±2.03 | 0.25±.03/0.07±.02 | 3.53±.48/5.37±1.16 | 0.01±.0 |
| Term2Note | 3768 | 0.39 | 0.40/0.13 | 0.47/1.14 | 0.53 |
| | 3364.43±118.37 | 0.39±.11 | 0.41±.0/0.13±.0 | 0.43±.01/1.14±.02 | 0.42±.13 |

*(Left margin label: Synthetic)*

Table 5: Supplementary results for fidelity evaluation: text length, term distribution, and semantic similarity (MAUVE). The **best result** among all methods to generate synthetic datasets is shown in bold, and the best result at the same privacy cost is underlined. Values in gray are aggregated across multiple inferences without DP quality maximiser applied.

olds. We then assess how well each metric identifies poor-quality sections using scalar thresholds. Our results indicate that metrics based on sentence length—particularly maximum sentence character count—align most closely with human annotations. A rejection threshold of 2181 characters yields the strongest correspondence. Table 6 reports the KL divergence and MAUVE values of synthetic notes after integrating this metric into the inference process. Specifically, if a generated section exceeds the threshold, it is discarded and regenerated until acceptance. Incorporating this simple criterion yields measurable improvements, suggesting that lightweight, reference-free filters can enhance the realism of DP synthetic text. Future work may extend this approach by combining multiple RF metrics for greater robustness.

| **Dataset** | **KL Divergence**↓ | **MAUVE**↑ |
|---|---|---|
| w/o RF metric | 1.99±0.37 | 0.24±0.04 |
| w/ RF metric | 1.03±0.10 | 0.36±0.07 |

Table 6: Fidelity evaluation of synthetic notes generated with and without integrating the RF metric (maximum sentence character count) into the inference process.

**Comparison under** $\epsilon = 8$   Figure 6 shows the distribution of sequence lengths for clinical notes generated by different methods under a fixed privacy budget of $\epsilon = 8$. While all synthetic methods shift the length distribution away from the original data to some extent, Term2Note exhibits the closest alignment. Its distribution captures the broad length range and multi-modal structure of the original notes more faithfully than the baselines. In contrast, AUG-PE produces much shorter and more narrowly distributed sequences, indicating a loss of structural richness. FastDP also generates relatively short sequences, with a sharp peak around 500 tokens. These deviations suggest that Term2Note is better able to preserve the structural properties of real clinical notes, which is crucial for downstream utility and realism in synthetic data.

## I.2   UTILITY

Table 7 presents the detailed precision and recall scores for the downstream task evaluation.

## I.3   GEMMA

Experimental results for Gemma are reported in Table 8. Both fidelity and utility metrics are comparable to those of Llama in Table 1, although the MAUVE score for Gemma without DP (i.e., $\epsilon = \infty$) is higher than that of Llama. Overall, the same trend holds across both models: stricter privacy guarantees lead to reduced fidelity, while full privatisation still preserves strong fidelity and utility.

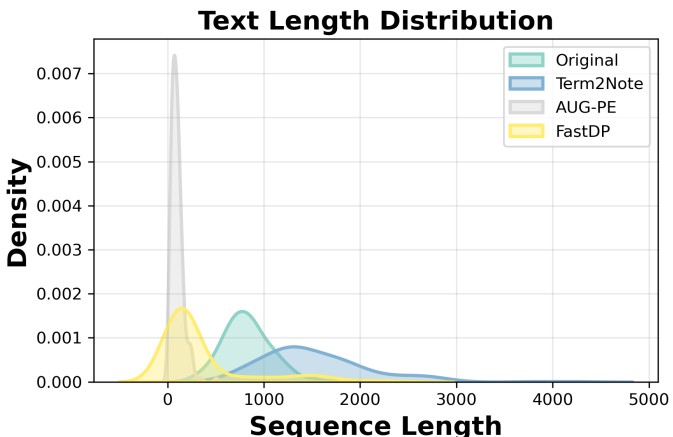

Figure 6: Text length distribution.

| Method | F1 | | Precision | | Recall | | AUC | | Precision@$k$ | |
|---|---|---|---|---|---|---|---|---|---|---|
| | Micro | Macro | Micro | Macro | Micro | Macro | Micro | Macro | $k=3$ | $k=5$ |
| Original Data | 57.03±3.59 | 30.80±2.20 | 60.14±4.51 | 34.66±4.56 | 54.40±4.26 | 30.88±2.66 | 82.01±1.36 | 58.88±2.73 | 68.93±4.53 | 62.14±3.20 |
| | | | | $\epsilon = \infty$ | | | | | | |
| AUG-PE | 45.82±2.33 | 14.84±2.97 | 67.30±5.10 | 17.91±5.33 | 34.94±3.19 | 16.27±2.67 | 79.52±1.23 | 54.35±1.75 | 68.48±6.11 | 61.77±3.03 |
| FastDP | 53.02±3.03 | 25.51±1.92 | 56.19±5.62 | 26.83±2.31 | 50.99±6.67 | 27.27±4.23 | 79.35±1.55 | 51.05±2.48 | 69.77±4.99 | 60.39±4.44 |
| Term2Note ($\epsilon_n = \infty$) | 49.95±4.77 | 21.89±3.90 | 65.37±6.31 | 28.69±1.26 | 41.02±6.64 | 21.05±3.86 | 81.40±1.78 | 55.43±2.40 | 69.77±3.52 | 61.96±5.70 |
| w. $\epsilon_t = \infty$ | 49.24±2.63 | 21.9±2.01 | 61.07±4.02 | 26.75±2.9 | 41.61±4.73 | 21.64±3.29 | 80.24±1.56 | 51.54±1.81 | 67.81±3.66 | 61.08±3.71 |
| | | | | $\epsilon = 8$ | | | | | | |
| AUG-PE | 40.73±9.22 | 13.28±4.71 | 60.68±4.70 | 15.14±4.77 | 31.43±9.80 | 16.01±5.81 | 78.13±1.85 | 53.49±3.08 | 63.24±4.65 | 59.03±4.59 |
| FastDP | 48.58±5.93 | 16.40±4.01 | 64.79±1.33 | 16.70±3.67 | 39.49±8.78 | 18.89±5.16 | 80.74±1.41 | 51.57±3.46 | 69.79±2.25 | 61.59±4.42 |
| Term2Note ($\epsilon_n = \infty$) | 49.71±1.90 | 21.28±1.28 | 61.37±4.28 | 25.17±1.55 | 41.96±2.98 | 21.31±1.63 | 80.03±1.44 | 52.80±1.15 | 67.49±4.42 | 61.48±4.45 |
| w. $\epsilon_t = \infty$ | 48.16±4.39 | 20.63±3.18 | 59.07±4.04 | 26.64±5.14 | 40.84±5.3 | 20.68±3.41 | 78.72±0.85 | 50.01±4.19 | 65.35±5.21 | 58.72±3.9 |
| w. $\epsilon_t = 8$ | 52.31±4.45 | 26.5±3.95 | 53.04±3.34 | 26.9±2.62 | 51.87±6.45 | 28.66±4.65 | 78.19±0.91 | 50.17±2.33 | 67.81±5.88 | 57.36±3.3 |
| w. $\epsilon_t = 5$ | 49.52±4.44 | 21.36±3.11 | 58.74±6.32 | 24.59±5.36 | 43.23±5.68 | 22.04±3.52 | 79.08±1.34 | 50.11±4.08 | 68.29±5.36 | 60.68±4.65 |
| w. $\epsilon_t = 2$ | 53.25±1.1 | 24.66±1.82 | 56.93±2.49 | 28.23±4.52 | 50.16±2.52 | 26.94±2.17 | 78.85±0.98 | 49.41±2.86 | 68.31±5.53 | 59.91±3.34 |
| | | | | $\epsilon = 5$ | | | | | | |
| AUG-PE | 48.10±2.08 | 17.44±2.70 | 60.59±8.29 | 18.28±3.69 | 40.73±5.59 | 21.02±4.88 | 77.78±3.11 | 53.33±2.60 | 63.01±12.42 | 56.94±5.20 |
| FastDP | 49.30±4.04 | 16.23±3.04 | 64.49±6.50 | 15.57±2.73 | 40.84±7.33 | 19.70±4.87 | 80.54±1.36 | 54.22±1.97 | 67.31±7.22 | 61.98±3.34 |
| Term2Note ($\epsilon_n = \infty$) | 47.94±4.47 | 20.31±3.40 | 60.88±4.95 | 25.76±4.44 | 40.15±6.22 | 20.20±3.71 | 79.29±1.49 | 51.19±3.40 | 66.04±5.73 | 61.69±4.86 |
| w. $\epsilon_t = \infty$ | 51.0±1.41 | 22.73±2.22 | 56.7±4.95 | 24.89±4.24 | 46.79±3.93 | 24.62±3.03 | 78.8±1.78 | 50.37±1.78 | 67.15±5.88 | 59.91±3.12 |
| w. $\epsilon_t = 5$ | 54.83±2.24 | 28.96±1.81 | 51.64±4.44 | 29.07±4.58 | 58.92±4.33 | 34.06±2.99 | 78.2±0.8 | 50.32±3.94 | 64.56±4.1 | 57.18±3.09 |
| w. $\epsilon_t = 2$ | 51.26±1.97 | 21.45±1.88 | 58.92±4.45 | 25.08±3.48 | 45.63±3.55 | 23.28±1.99 | 79.05±1.37 | 50.44±3.5 | 66.36±5.89 | 60.49±4.89 |
| | | | | $\epsilon = 2$ | | | | | | |
| AUG-PE | 40.9±7.43 | 13.57±3.88 | 64.75±7.31 | 15.2±4.82 | 30.7±9.22 | 14.83±5.29 | 78.29±0.8 | 53.38±1.79 | 63.74±5.0 | 60.1±4.37 |
| FastDP | 51.06±5.7 | 20.04±4.34 | 58.78±5.13 | 19.77±4.27 | 45.94±8.57 | 23.72±5.52 | 79.98±1.95 | 51.31±3.32 | 66.32±6.66 | 59.99±5.35 |
| Term2Note ($\epsilon_n = \infty$) | 51.78±3.99 | 3.36±3.87 | 57.45±5.97 | 25.59±4.84 | 47.21±3.16 | 25.26±2.64 | 79.00±1.67 | 50.60±2.75 | 67.00±3.39 | 59.52±5.59 |
| w. $\epsilon_t = \infty$ | 48.57±0.89 | 20.31±1.25 | 59.92±5.19 | 24.63±2.69 | 41.23±3.46 | 20.75±2.05 | 79.56±1.08 | 51.75±0.85 | 68.64±3.55 | 60.41±3.43 |
| w. $\epsilon_t = 2$ | 51.87±2.73 | 23.06±2.51 | 57.77±6.18 | 26.58±5.04 | 47.37±2.88 | 24.77±2.01 | 79.43±1.41 | 51.3±2.96 | 69.45±4.73 | 60.31±5.39 |

_(Rows below $\epsilon = \infty$ through $\epsilon = 2$ are grouped under the label **Synthetic**.)_

Table 7: Supplementary results for utility evaluation: F1, Precision, Recall, AUC, and Precision@$k$, with **mean±standard deviation** values reported.

## J  CASE STUDIES FOR TERM GENERATION

Table 9 presents examples of synthetic clinical terms generated with and without the application of DPRP*. The original list contains five salient terms extracted from a real clinical note. When no DP is applied, the generated list recovers only two of these terms ("air" and "discharge"), suggesting limited coverage despite the absence of privacy constraints. In contrast, the DP-enabled output does not directly replicate any of the original terms beyond "discharge", but instead generates a substantially longer and more diverse list of medically plausible terms.

This illustrates a key trade-off: the DP mechanism introduces sufficient variability to obscure direct term recovery, thus enhancing privacy protection. At the same time, the generated list remains semantically coherent and clinically relevant, containing realistic phrases such as "hemodynamically stable," "chronic low back pain," and "pulmonary vein," which contribute to the naturalness and

| Method | Fidelity | | | | Utility | | | | | |
|---|---|---|---|---|---|---|---|---|---|---|
| | **Length** | **Unary/Binary Term** | | **Semantic** | **F1** | | **AUC** | | **Precision@$k$** | |
| | KL Div.↓ | Jaccard↑ | KL Div.↓ | MAUVE↑ | Micro | Macro | Micro | Macro | $k=3$ | $k=5$ |
| Original Data | | | | | 57.03 | 30.80 | 82.01 | 58.88 | 68.93 | 62.14 |
| $\epsilon = \epsilon_n = \infty$ | | | | | | | | | | |
| Term2Note | **0.18** | **0.61/0.34** | **0.23**/1.55 | **0.80** | 52.56 | 25.32 | **80.81** | **55.09** | 67.30 | 60.78 |
| w. $\epsilon_t = \infty$ | 0.36 | 0.40/0.18 | 0.77/**1.52** | 0.66 | 53.77 | 25.72 | 79.94 | 51.86 | 66.98 | 60.71 |
| $\epsilon = \epsilon_n = 8$ | | | | | | | | | | |
| Term2Note | 0.40 | 0.39/0.13 | 0.53/1.66 | 0.48 | 48.41 | 20.89 | 78.55 | 48.89 | 67.50 | 59.31 |
| w. $\epsilon_t = \infty$ | 0.40 | 0.37/0.13 | 0.64/1.82 | 0.38 | 47.63 | 20.19 | 78.87 | 50.5 | 66.83 | 59.53 |
| w. $\epsilon_t = 8$ | 0.40 | 0.37/0.13 | 0.75/2.05 | 0.32 | **55.46** | 26.73 | 79.68 | 49.92 | 69.30 | 60.60 |
| w. $\epsilon_t = 5$ | 0.41 | 0.38/0.13 | 0.70/1.71 | 0.38 | 51.82 | 21.67 | 80.44 | 51.28 | **70.92** | 60.88 |
| w. $\epsilon_t = 2$ | 0.51 | 0.36/0.12 | 0.76/1.89 | 0.30 | 52.34 | 22.50 | 80.02 | 51.71 | 69.62 | 60.01 |
| $\epsilon = \epsilon_n = 5$ | | | | | | | | | | |
| Term2Note | 0.36 | 0.38/0.13 | 0.56/**1.52** | 0.31 | 51.20 | 21.18 | 80.03 | 53.10 | 67.65 | **61.38** |
| w. $\epsilon_t = \infty$ | 0.25 | 0.37/0.13 | 0.65/1.76 | 0.32 | 49.98 | 21.50 | 79.77 | 52.57 | 68.14 | 59.91 |
| w. $\epsilon_t = 5$ | 0.39 | 0.36/0.12 | 0.75/1.79 | 0.32 | 52.79 | 23.74 | 79.54 | 49.64 | 67.99 | 58.92 |
| w. $\epsilon_t = 2$ | 0.41 | 0.36/0.13 | 0.77/1.83 | 0.21 | 55.06 | **27.86** | 79.42 | 49.19 | 69.30 | 59.82 |
| $\epsilon = \epsilon_n = 2$ | | | | | | | | | | |
| Term2Note | 0.49 | 0.36/0.12 | 0.60/1.65 | 0.27 | 49.26 | 21.17 | 79.21 | 50.33 | 67.16 | 59.24 |
| w. $\epsilon_t = \infty$ | 0.31 | 0.36/0.13 | 0.69/1.59 | 0.35 | 48.11 | 20.87 | 79.49 | 52.83 | 65.51 | 57.17 |
| w. $\epsilon_t = 2$ | 0.46 | 0.35/0.12 | 0.80/1.78 | 0.31 | 53.66 | 24.43 | 79.91 | 50.26 | 69.28 | 61.08 |

(All Term2Note rows are under the row group labelled "Synthetic".)

Table 8: Fidelity and utility evaluation of synthetic datasets generated by Term2Note with Gemma-3-1B as the base model for NOTEGEN.

utility of the resulting synthetic note. These findings align with our earlier quantitative analysis, confirming that DPRP* balances semantic fidelity with privacy-preserving diversity.

| Method | Terms |
|---|---|
| Original | ["Physical", "Discharge", "Laparoscopic", "incisions", "air"] |
| No DP | ["air", "discharge"] |
| DP | ["brief", "discharge", "negative", "medications", "placement", "drainage", "hemodynamically stable", "therapy", "chronic low back pain", "symptoms", "right chest", "referred to cardiac surgery", "chest discomfort", "pulmonary vein", "hyperlipidemia: he", "hypertension-", "difficulty", "surgical service", "anticoagulation", "discontinued", "increased", "afebrile", "asymptomatic", "admission", "intervention", "hospitalization", "cardiac enzymes x3"] |

Table 9: Example of synthetic terms.

# K    EXAMPLE DEMONSTRATION

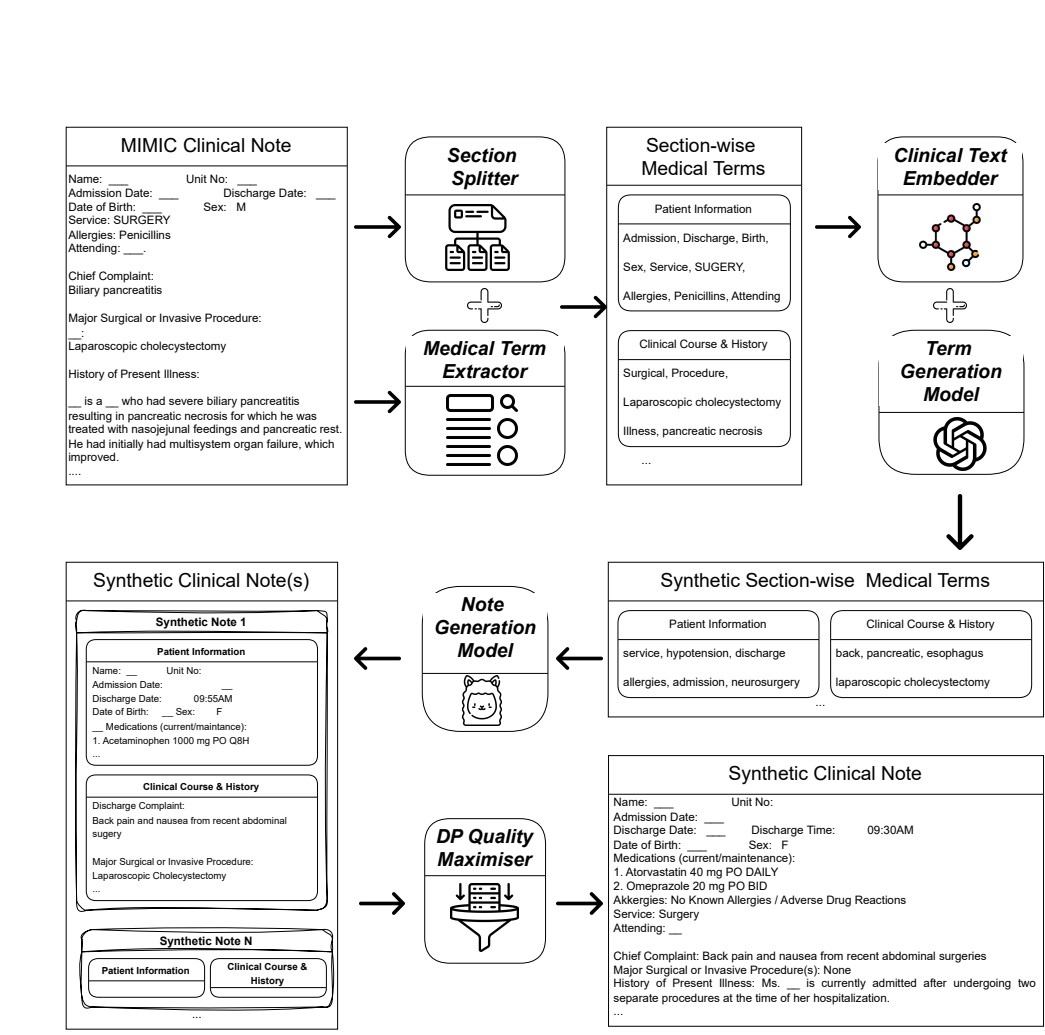

Figure 7: Example to demonstrate Term2Note.

