# OpenReview forum: "Term2Note: Synthesising  Differentially Private Clinical Notes from Medical Terms"
_ICLR.cc/2026/Conference — ICLR 2026 Conference Withdrawn Submission_

### Official Review · Reviewer_PvSU · 2025-11-01

**Soundness:** 1
**Presentation:** 2
**Contribution:** 2
**Rating:** 2
**Confidence:** 5

**Summary:**

The paper presents Term2Note, a framework for generating differentially private synthetic clinical notes from medical terms. It addresses the challenge of creating long-form medical text that ensures both privacy protection and data utility. Term2Note generates clinical notes section by section. Each section is based on the extracted medical terms and the previously generated sections, and each process, term generation and note generation, is protected under independent differential privacy (DP) constraints. The framework further enhances text quality through a DP quality maximiser, which selects the most coherent outputs based on perplexity. The authors claim that Term2Note achieves high fidelity to real clinical notes, competitive utility in downstream ICD code prediction tasks, and superior performance to existing DP text generation baselines, while maintaining formal privacy guarantees.

**Strengths:**

The paper proposes a new structured framework for generating long-form synthetic clinical text, a problem that has rarely been explored due to the complexity of medical contents and strict privacy constraints. The framework is systematically modularized, dividing the synthetic data generation process into distinct stages—term extraction, term generation, note generation, and quality maximization—which makes the overall structure organized and understandable. It also effectively leverages existing DP algorithms and their theoretical properties to ensure formal privacy guarantees throughout the workflow, maintaining a careful balance between privacy and utility. Overall, the work holds methodological significance for healthcare AI by suggesting a new approach to generating and sharing sensitive medical data, addressing one of the major challenges in privacy-preserving data acquisition.

**Weaknesses:**

1. Limited Stability of Utility Results

The paper claims that Term2Note consistently outperforms other baselines in terms of fidelity and utility. This claim is well supported for fidelity, as Term2Note shows superior structural, syntactic, and semantic similarity across all privacy settings. However, the result in utility do not fully support this statement. While Term2Note achieves the highest F1 score, some results in AUC and Precision values are slightly higher for AUG-PE and FastDP across privacy budgets (ε = 8, 5, and 2), according to Table 1. It suggests that the advantage in utility is not fully uniform across settings. Also, the inconsistency between statement and experiment results can weaken the credibility of the paper. Acknowledging these variations and discussing their possible causes, such as model randomness or task sensitivity, would make the analysis more balanced and credible.

2. Concerns Raised in Qualitative Analysis

Physicians’ feedback in Qualitative Analysis section suggests discrepancies with the paper’s assertion of high-quality text generation. The evaluators pointed out several issues in Term2Note outputs, including missing or misordered sections, internal inconsistencies, and vague or generic phrasing. One physician also noted medication misclassification, illogical or irrelevant narrative insertions, and factual errors. Although these issues may not be prevalent, their presence indicates that further refinement is needed to ensure clinical reliability and practical usability. The authors are encouraged to investigate these qualitative errors systematically and discuss possible mitigation strategies.

3. Lack of Clarity in Notation and Experimental Setting

The paper omits several explanations that are necessary for readers to fully understand the framework. In 2 Background & Related Work, the mathematical formulation of DP is insufficiently explained: variables such as 𝜒 and 𝑅 appear without definition, making the DP derivation difficult to follow. This paper replaces empirical validation of the crucial element of privacy protection with mathematical proofs alone, without sufficient experiments. Therefore, a highly rigorous mathematical statement is required.
Moreover, in 4 Experiment Setting, the baseline description is uneven: while the DP-SGD methodology underlying FastDP is explained earlier in the paper, AUG-PE is only briefly mentioned in one sentence without sufficient explanation of its mechanism or how it differs from Term2Note. Providing explicit definitions and more balanced baseline explanations would greatly improve readability.

4. Insufficient Justification for Model and Hyperparameter Choices

The paper lacks justification for model selection and hyperparameter configuration. Term2Note integrates multiple large models—GPT-2 Large, Llama-3.2-1B or Gemma-3-1B, Asclepius-Llama3-8B—but the rationale for choosing these specific architectures or parameter scales is not discussed. In addition, the paper sets the ε (epsilon) hyperparameters to 2, 5, and 8 for both term-level and note-level privacy budgets, yet provides no explanation for why these specific values were chosen or how they correspond to concrete levels of privacy protection. A short discussion of how these ε values translate to practical privacy guarantees would help readers better interpret the trade-off between privacy and utility.

**Questions:**

1. Scope of DP Definition: In 2 Background & Related Work, you define DP for an arbitrary randomized function. Immediately after, the text states “any function”. Is this extension intentional and mathematically valid? If so, please clarify the conditions under which the result holds, and cite a supporting theorem. Also, please provide the definition for “randomized function” or “randomized algorithm”.

2. TERMEXT Notation Consistency (Sec. 3.2): The final line in 3.2 Format and Term Identification writes TERMEXT([SEC1, ... , SECm]). Should this instead write [TERMEXT(SEC1), ... , TERMEXT(SECm)]? If the current form is correct, please explain how TERMEXT operates.

3. Index Variable Mismatch (Alg. 1 vs. Sec. 3.4): Algorithm 1 says 𝑗 indexes sections, while Sec. 3.4 uses 𝑖 for sections. Is one of these a typo? Please standardize the index variable across the paper or state the mapping explicitly.

4. Model Capacity for Note Generation: Physician feedback points to missing/misordered sections, internal inconsistencies, and medication errors. Consider evaluating a larger NOTEGEN (e.g., next-scale Llama/Gemma) under the same DP budgets. It could show whether quality deficits stem from model size rather than the framework.

5. End-to-End Example Through the Paper: To make the framework easier to follow, it would be helpful to include one representative example that runs through the entire Term2Note pipeline. Such an example would help readers intuitively understand how the original text is transformed into a synthetic note and how potential re-identification risks are mitigated.

6. Paragraph Break in Sec. 3.3 (Line 190): Up to “projected embeddings.” appears to explain Algorithm 1, Line 5; the text then shifts to Line 7. It would be better to split into two paragraphs or add a transition sentence to mark the change in step.

7. Clarification on ε Bound (Sec. 3.6): In Section 3.6, the paper states that the overall ε is upper-bounded by the larger of εₙ and εₜ. However, in Table 1, configurations are presented where εₜ = ∞ while εₙ decreases. If ε is bounded by the larger value, the overall privacy budget should remain infinite. How is this reconciled? Please provide an explanation.

8. Explanation and examples for MIMIC Datasets: The paper uses MIMIC-III as the public dataset and MIMIC-IV as the private dataset. Could the authors elaborate on the key differences between the two and justify why each is suited for its respective role? Were any preprocessing or normalization steps? Providing an example or brief explanation would help readers understand the rationale for this.

9. Clarification on Semantic Similarity Evaluation: The paper equates semantic similarity with distributional alignment measured in MAUVE. However, MAUVE evaluates global distributional fidelity rather than sentence- or concept-level semantic similarity. Could you clarify how distributional similarity reflects semantic similarity, or justify this interpretation?

10. Typos and Style Consistency:

- Line 223: “preform” → “perform.”

- Line 485: Replace the period after “NLP” with a comma.

- Table 1: Unify the spelling of FastDP (with or without hyphen) across the paper.

---

> ### Author Response · Authors · 2025-11-26
> **Response to Reviewer PvSU - Part 1**
>
> We thank the reviewer for the thoughtful feedback. Below, we address each of the points raised in detail.
>
> ### Weakness
>
> ***Weakness 1: limited stability of utility results***
>
> F1, AUC, and Precision@k capture different aspects of model performance: F1 reflects threshold-dependent classification accuracy, AUC measures ranking ability across all thresholds, and Precision@k evaluates correctness among the top-k predictions. Because these metrics emphasise different behaviours, it is expected that improvements may appear stronger on some metrics than others, especially given the difficulty of the downstream task (a 20-label classification problem) and the limited amount of training data (< 200 samples). For instance, even the original data shows a lower Precision@3 score than some synthetic datasets, which reflects task difficulty rather than shortcomings of the real data. Therefore, the observed variation across metrics reflects the multifaceted nature of utility evaluation under a challenging setting, rather than instability in Term2Note.
>
> Importantly, when comparing synthetic-data performance against the original dataset, Term2Note achieves the closest performance in F1 and delivers comparable performance to the baselines in AUC and Precision@k. This demonstrates that Term2Note preserves clinical utility effectively. We have revised the discussion in the updated submission to explicitly acknowledge these variations and provide a more balanced interpretation of the results.
>
>
> ***Weakness 2: concerns raised in qualitative analysis***
>
> We would like to offer several clarifications and additional context regarding the qualitative analysis conducted with physicians.
>
> (1) This qualitative analysis was explicitly designed to **elicit any potential clinical issues** in the synthetic notes. Physicians were asked to critically inspect the outputs and flag problems. Under this evaluation setup, negative findings are expected and indeed valuable. We have clarified this intention more explicitly in the revised submission to avoid misinterpretation.
>
> (2) **Not all notes contain issues**. Among the 20 notes reviewed by one physician, 9 (45%) were assessed as having no clinical issues. This proportion reflects that while imperfections remain, Term2Note is capable of producing clinically plausible notes in nearly half of the sampled notes, which is an encouraging outcome given the difficulty of long-form clinical note generation.
>
> (3) **Contextualising the qualitative issues**: Hallucinations and inconsistencies are common challenges in clinical text generation, not unique to Term2Note. Importantly, despite these issues, physicians consistently preferred Term2Note over the baseline (Figure 3), indicating that Term2Note produces comparatively more coherent and clinically meaningful notes. Thus, the qualitative feedback both highlights areas for improvement and underscores Term2Note’s relative strengths.
>
> In the updated submission, we have expanded the discussion to acknowledge these limitations more explicitly and to outline potential mitigation strategies, such as integrating clinical consistency checks into the generation pipeline.
>
>
> ***Weakness 3: lack of clarity in notation and experimental setting***
>
> Specifically, $\mathcal{X}$ denotes the domain of possible records, and $R$ denotes the output space of the randomised mechanism.
>
> We respectfully clarify that our work does not intend to replace empirical privacy evaluation with mathematical proofs alone. Instead, we adopt the standard formal DP guarantee as the basis for privacy protection, similar to other DP baselines. Since DP provides a worst-case guarantee that holds for any adversary, empirical privacy attacks are not required to validate the mechanism’s DP property. Nonetheless, we acknowledge the reviewer’s point that demonstrating empirical robustness could further strengthen the paper; we plan to incorporate such evaluations, such as membership inference attacks, to complement the theoretical protection.
>
> Finally, we have made the revisions for both notation and baseline explanation in the updated submission.

---

> ### Author Response · Authors · 2025-11-26
> **Response to Reviewer PvSU - Part 2**
>
> ***Weakness 4: insufficient justification for model and hyperparameters choices***
>
> The choice of GPT-2 as the backbone term-generation model and Asclepius-Llama3-8B as the perplexity model is already explained in *Section 3.7 Implementation Details – Backbone Models*. Our choice of Llama-3.2-1B/Gemma-3-1B as backbone generators was guided by two considerations: (1) both models were among the most widely adopted open-source LLMs available at the time of experimentation, providing strong community support and ensuring that our evaluation of Term2Note remains reproducible; and (2) we explicitly sought model diversity, allowing us to assess Term2Note’s robustness across distinct model families. The 1B parameter scale was chosen due to computational constraints inherent to DP fine-tuning. Training DP models on 120k clinical notes with sequence length in average 3.3k tokens incurs a substantial overhead from gradient clipping and per-sample noise injection. Even with DeepSpeed ZeRO-3 and a batch size of 2, a single DP training run required approximately 2-3 days on two 80GB A100 GPUs. Given these constraints, scaling to larger LLMs was not feasible with our available resources. We have clarified this rationale in the revised submission.
>
> Regarding the choice of $\epsilon$ values, we adopt the values following common practice in prior DP work (e.g., in [1], the authors also use 2, 5, 8), where $\epsilon=2$ corresponds to high-privacy; $\epsilon=5$ corresponds to moderate-privacy; and $\epsilon=8$ reflects a more utility-oriented configuration.
>
> [1] Da Yu, Huishuai Zhang, Wei Chen and Tie-Yan Liu, [Do not Let Privacy Overbill Utility: Gradient Embedding Perturbation for Private Learning](https://arxiv.org/abs/2102.12677), ICLR, May 2021
>
> ---
> ### Questions
> ***Question 1: DP definition***
>
> The phrase "any function" was intended specifically in the context of post-processing of DP, not as a statement that arbitrary functions are themselves DP. We have revised the paragraph in Section 2 to clarify that the post-processing theorem applies to any deterministic or randomised function applied to the output of a DP mechanism.
>
> A randomised algorithm (or randomised function) $\mathcal{M}: \mathbb{N}^{|\mathcal{X}|} \to \mathcal{R}$ maps a dataset to a probability distribution over outputs, where the output of $\mathcal{M}(D)$ depends on internal randomness.
>
>
> ***Question 2: notation consistency in section 3.2***
>
> Thank you for pointing out this inconsistency. In the revised submission, we now explicitly write $[T_1^{src}, ..., T_m^{src}] = [TermExt(SEC_1^{src}), ..., TermExt(SEC_m^{src})]$, which makes clear that TermExt is applied independently to each section, rather than to a list as a whole.
>
> ***Question 3: index variable mismatch in algorithm 1 and section 3.4***
>
> In the revised submission, we have standardised the notation by using the same index variable throughout.
>
> ***Question 4: larger LLMs***
>
> As noted in our response to Weakness 4, training NoteGen under DP-SGD is computationally demanding due to the combination of (1) long-context inputs, (2) a training set exceeding 120k instances, and (3) the substantial overhead introduced by per-sample gradient clipping and noise addition. Even with DeepSpeed ZeRO-3 and a micro-batch size of 2, training a 1B-parameter model already requires 2–3 days on two 80GB A100 GPUs. Scaling to larger Llama/Gemma variants (e.g., 3B–8B) would exceed our available compute resources.
>
> While parameter-efficient fine-tuning methods such as LoRA could in principle reduce training costs, two practical challenges remain: (1) our preliminary non-private experimentation using QLoRA on Llama-3.3-70b 4-bit with only 10k training instances (presented in Appendix I, Table 5) still required ~80 hours of training; and (2) integrating LoRA with DP-SGD is non-trivial because it requires correctly implementing per-sample gradient computation and noise addition for the adapted parameters.
>
> Thus, while we were unable to include larger NoteGen models in this submission, we agree that increased model capacity may help address some qualitative issues highlighted by physicians (e.g., inconsistencies and medication errors). As part of our future work, we plan to explore DP with LoRA to enable training larger NoteGen backbones under the same privacy budgets. This will help disentangle the effect of model capacity from the Term2Note framework itself.
>
> ***Question 7: $\epsilon$ clarification***
>
> We apologise for the confusion. Upon re-examining the privacy accounting, we confirm that the overall privacy budget indeed becomes $\infty$ whenever either $\epsilon_t = \infty$ or $\epsilon_n = \infty$, since in both cases at least one module accesses private data without DP protection. We have corrected Table 1 and updated the corresponding discussion in the revised submission to accurately reflect this fact.

---

> ### Author Response · Authors · 2025-11-26
> **Response to Reviewer PvSU - Part 3**
>
> ***Question 8: explanation and examples for MIMIC datasets***
>
> MIMIC-III and MIMIC-IV are both compiled by the MIT research group, with MIMIC-IV designed "to carry on the success of MIMIC-III and support a broad set of applications within healthcare"[2]. MIMIC-IV is larger, includes more recent clinical data, and offers improved note formatting. For these reasons, we treat MIMIC-IV as the “private” dataset, representing the realistic setting in which a hospital has access to a large, sensitive clinical note corpus that must be protected.
>
> MIMIC-III has long served as a standard public benchmark for clinical NLP research. In our framework, it provides auxiliary public clinical terms. While MIMIC-III is not uniquely required (any publicly available dataset containing clinical terms could serve this role), it is a natural and accessible choice.
>
> Regarding preprocessing, both datasets are minimally modified. Notes are used in their original form except for the standard steps described in Section 3.2. No additional normalisation is applied, ensuring that the natural structure of clinical documentation is preserved for meaningful fidelity evaluation.
>
> [2] Johnson, A., Bulgarelli, L., Pollard, T., Gow, B., Moody, B., Horng, S., Celi, L. A., & Mark, R. (2024). MIMIC-IV (version 3.1). PhysioNet. RRID:SCR_007345. https://doi.org/10.13026/kpb9-mt58
>
> ***Question 9: clarification on MAUVE***
>
> We agree that MAUVE is not a sentence-level semantic similarity metric like BERTScore or cosine similarity between embeddings. As noted in the paper, MAUVE measures the divergence between two global distributions of text by quantising pretrained language model embeddings. While this is fundamentally a distributional metric, the embedding space it relies on is known to encode high-level semantic and stylistic information. For this reason, we interpret MAUVE as reflecting semantic alignment at the distribution level, rather than claiming it captures fine-grained sentence semantics. We have made this distinction clearer in the revised submission.
>
> Importantly, sentence-level semantic metrics are less appropriate for evaluating DP-generated text. By definition, DP mechanisms intentionally break one-to-one correspondence between real and synthetic instances; thus, evaluating pairwise sentence similarity is not meaningful and may even conflict with privacy goals. Distributional-level measures such as MAUVE are better suited for assessing quality in this setting because they capture how well the overall characteristics of the synthetic text align with the real data distribution.
>
> For completeness, we experimented with a true sentence-level semantic measure (BERTScore). However, we found that BERTScore was not informative in this setting: scores were consistently above 0.94 across all methods, failing to differentiate synthetic outputs meaningfully. This supports the use of a distribution-level metric such as MAUVE, which better captures differences in overall textual behaviour for long clinical notes.
>
> ***Questions 5, 6, 10: end-to-end example, paragraph break, and typos***
>
> We have made the corresponding revisions as suggested in the updated submission. An end-to-end example is provided in Appendix K.

---

### Official Review · Reviewer_z7pN · 2025-11-02

**Soundness:** 2
**Presentation:** 3
**Contribution:** 2
**Rating:** 4
**Confidence:** 3

**Summary:**

This paper focuses on how to acquire differentially private clinical notes for reliable, privacy-preserved research. The paper proposes Term2Note, a method that first produces a series of medical terms, then generates private clinical notes based on these terms. To achieve this, Term2Note includes a fine-tuned embedder and a TermGen model for term generation, whose objective is to reconstruct the ground-truth term lists. Next, Term2Note includes a NoteGen model responsible for generating clinical notes. Experiments demonstrate the fidelity (measured by how closely the generated notes resemble real data) and utility (measured by their usefulness for downstream tasks).

**Strengths:**

1. The design of term generation followed by text generation makes the model more transparent and explainable, which could be essential for clinical use.
2. Experimental analyses are conducted under various $\epsilon$ settings, which helps reveal the applicability of the proposed method under different privacy requirements.
3. Compared with automated metrics, human evaluation with licensed physicians makes the results more convincing.

**Weaknesses:**

1. Some ablation studies could demonstrate the role of each module more clearly. For example, according to Algorithm 1, Term2Note first relies on $\theta_t$ and $\theta_p$ to generate clinical terms, and the training objective is to reconstruct the ground-truth terms in $D_{public}$. As an ablation, what if the model directly uses the ground-truth terms from $D_{public}$ during inference? What would the performance gap be, and how would the term generation model perform? This becomes more important since $\theta_t$ is GPT-2-Large, which does not have very strong capability compared with current LLMs.
2. The experimental results do not show solid improvements. According to Table 1, the utility measures between Term2Note and the baselines are comparable, and FastDP often performs better across various $\epsilon$ settings. This seems to contradict the analysis that Term2Note achieves the best performance under varying privacy budgets (L421–423).
3. Some implementation details could be described more clearly. For example, term generation first projects the embeddings with $\theta_p$, then generates using the generative model $\theta_t$. How are the embeddings provided to $\theta_t$ (e.g., fed as inputs, used in cross-attention during generation, etc.)?
4. The main contribution of this paper seems to lie in the pipeline design (first generate medical terms, then generate clinical notes). The algorithms used, such as FastDP, are from previous works. Thus, the technical contribution could be limited.
5. Minor: Figure 2 includes negative N-gram frequency. Although this is due to smoothing, negative N-gram frequency seems do not have physical meanings, and maybe the paper should consider truncate them.

**Questions:**

Please see the weakness, especially for 1-4. The rebuttal could focus on better explanation of technical innovation and focus on some ablation studies to demonstrate the role of term generation and notes generation respectively. Also, It could be helpful to explain the experiment results and implementation clearer (Weakness 2,3) or highlights the part I misunderstood.

---

> ### Author Response · Authors · 2025-11-26
> **Response to Reviewer z7pN**
>
> We appreciate the reviewer’s insightful comments. We provide point-by-point responses below .
>
> ---
> ***Weakness 1: ablation analysis***
>
> Thank you for your suggestion. We would like to clarify that the ablation analysis about using ground-truth clinical terms instead of synthetic terms during inference is reflected in Table 1 by the rows labelled Term2Note ($\epsilon_n=...$), i.e., the configuration where $\epsilon_t$ is not applied. In these settings, the pipeline bypasses the TermGen module entirely and uses the original term lists extracted from the private notes, i.e., ground-truth terms.
>
> As expected, notes generated from ground-truth terms achieve higher fidelity scores compared to those generated from synthetic terms, though the improvement is not dramatic. To quantify TermGen’s reconstruction capability, we evaluated how well GPT-2–based TermGen recovers the original term lists. It achieves an F1 score of 0.37 and an embedding cosine similarity of 0.82 relative to the ground-truth. This indicates moderate reconstruction accuracy while preserving semantic alignment.
>
> Importantly, even when using synthetic terms, Term2Note outperforms baseline methods in fidelity and achieves comparable utility scores, demonstrating the effectiveness of the TermGen module despite the modest reconstruction capability.
>
> ***Weakness 2: improvements are not solid***
>
> We appreciate the reviewer’s observation. The utility metrics, F1, AUC, and Precision@k, capture different aspects of model performance. F1 reflects threshold-dependent classification accuracy, AUC measures ranking ability across all thresholds, and Precision@k evaluates correctness among the top-k predictions. Because these metrics emphasise different behaviours, it is expected that improvements may appear stronger on some metrics than others, especially given the difficulty of the downstream task (a 20-label classification problem) and the limited amount of training data (< 200 samples). For instance, even the original data shows a lower Precision@3 score than some synthetic datasets, which reflects task difficulty rather than shortcomings of the real data. Therefore, the observed variation across metrics reflects the multifaceted nature of utility evaluation under a challenging setting, rather than instability in Term2Note.
>
> Importantly, when comparing synthetic-data performance against the original dataset, Term2Note achieves the closest performance in F1 and delivers comparable performance to the baselines in AUC and Precision@k. This demonstrates that Term2Note preserves clinical utility effectively. We have revised the discussion in the updated submission to explicitly acknowledge these variations and provide a more balanced interpretation of the results.
>
> ***Weakness 3: unclear implementation details***
>
> Thank you for pointing this out. We have clarified this in the revised manuscript. Briefly, the term embeddings are first passed through a trainable projection layer to match the hidden dimensionality of the generative model. The projected embeddings are then provided to the model as its input sequence embeddings, replacing the usual token embeddings. In other words, the generative model conditions directly on these projected embeddings at the input layer and auto-regressively reconstructs the clinical term list. No cross-attention module is introduced; the conditioning is achieved entirely through the input embedding pathway.
>
> ***Weakness 4: limited technical contribution***
>
> We respectfully argue that our contribution goes beyond assembling a pipeline. Term2Note addresses two core technical challenges that prior DP text generation work has not resolved.
>
> (1) Term2Note enables DP generation of long, structured clinical notes with realistic section-level organisation, while existing DP text generation methods primarily operate at short text or paragraph scale and do not preserve long structure workflows.
>
> (2) Term2Note removes the need of prior knowledge of class or label distributions, while many prior DP text generation methods rely on access to global distribution. This allows flexible synthesis across heterogeneous note types and reduces leakage pathways associated with distributional priors.
>
> Together, these contributions make Term2Note more than a combination of existing components, but a new, effective approach that makes long-form, privacy-preserving clinical text generation practical and clinically meaningful.
>
>
> ***Weakness 5: negative n-gram frequency in Figure 2***
>
> Thank you for the suggestion. The negative values do arise from the smoothing procedure, and we kept them to present the full distribution for completeness. However, we agree that negative frequencies do not have a direct physical interpretation. We will consider truncating the values to zero in future versions.

---

### Official Review · Reviewer_RmfR · 2025-11-06

**Soundness:** 3
**Presentation:** 2
**Contribution:** 3
**Rating:** 2
**Confidence:** 4

**Summary:**

The paper introduces a novel framework named Term2Note for synthetic clinical note under DP constraint. The method process long-form clinical note by separating it into two sub tasks: key medical term extraction and note format generation. After applying DP training methods, experiment outcomes imply that Term2Note outperforms baseline models on fidelity to real clinical data, downstream task utility and human expert preference, while providing formal privacy guarantees. The author conducted thorough experiments on variants including base model and privacy budget, which provides a solid foundation for the main contribution.

**Strengths:**

The framework fulfills existing gap between synthetic text generation and differential private on clinical notes.

The evaluation is thorough。

**Weaknesses:**

The paper adopts FastDP but provides little explanation for why this particular method is preferred over other differentially private mechanisms (e.g., DP-SGD variants, PATE, or Rényi-based approaches). The DP section reads more like a straightforward application of an existing framework rather than a carefully motivated design.

The composition analysis is overly simplified and lacks a solid connection to established DP accounting techniques. From a differential privacy standpoint, the reasoning behind the overall privacy budget allocation and composition strategy is not well articulated, making the DP guarantee less convincing.

The empirical results in Table 2 appear inconsistent with the expected DP–utility trade-off. Specifically, when epsilon goes infinity the results should approximate the non-private baseline, and as ϵ decreases, utility should generally drop. However, this trend is not observed, and in some cases, DP models even outperform the non-DP setting, which is counterintuitive and suggests potential issues in experimental implementation or evaluation. And the budget is relatively large, especially after naive composition,  makes the dp guarantee very weak.

The paper does not clearly justify the use of the term extraction–narrative generation framework, nor discuss the mechanisms underlying this design. A brief explanation of the intuition behind this two-stage formulation (why term extraction benefits note generation and how it differs from end-to-end models) would help clarify the methodological motivation and strengthen the paper’s contribution.

The choice of baselines is not well justified. From a utility perspective, the paper should compare against clinical note generation methods without DP, even if they lack formal privacy guarantees. Within the DP setting, using only a simple DP-SGD + LLM configuration is not an adequate baseline for evaluating the effectiveness of FastDP.

The diagram illustrating the proposed framework lacks clarity—some arrows and dependencies are ambiguous, and the role of DP within the overall pipeline is not clearly reflected. The figure does not effectively communicate how the DP mechanism is integrated into the model training or generation process.

Comparing text length distribution with KL divergence do not intuitively capture structural or semantic similarity between the synthetic clinical notes and the real data. While this metric may reflect differences in overall length patterns, it provides limited insight into whether the generated notes preserve linguistic coherence or structural arrangement.

**Questions:**

As above in weakness

---

> ### Author Response · Authors · 2025-11-26
> **Response to Reviewer RmfR - Part 1**
>
> We thank the reviewer for engaging closely with our work. We address each point below and clarify the technical foundations of our approach.
>
> ---
> ***Weakness 1: little explanation about why FastDP***
>
> We respectfully disagree with the concern and clarify the rationale below.
>
> (1) **FastDP directly provides the DP-SGD + Rényi-DP accounting framework.** FastDP is not an alternative to DP-SGD or RDP; instead, it implements DP-SGD and supports RDP as its primary accounting method. This is exactly the mechanism used in our experiments. FastDP [1] is a library that allows differentially private optimisation of PyTorch models, supporting all PyTorch optimisers (e.g., DP-SGD) and multiple privacy accountants (e.g., Rényi-DP). Our choice of FastDP is primarily motivated by its computational and memory efficiency for large model fine-tuning.
>
> (2) Term2Note is DP-mechanism-agnostic; it does not rely on any FastDP-specific assumptions. Any DP variant for deep learning model training that provides equivalent formal guarantees can be substituted.  In this sense, FastDP serves as a practical, drop-in implementation choice rather than a core algorithmic dependency.
>
> [1] https://github.com/awslabs/fast-differential-privacy
>
> ***Weakness 2: overly simplified analysis about DP guarantee***
>
> We respectfully disagree with the assessment that our composition analysis is "overly simplified".
>
> A complete proof about the composition analysis is provided in the submission (Equation (4) and Appendix C). The critique does not identify any specific mathematical error or missing step, which makes it difficult to address concrete concerns. Nonetheless, we clarify the core reasoning below.
>
> In Term2Note, the overall privacy guarantee follows directly from the standard parallel composition theorem in DP. No explicit “allocation” of privacy budget between TermGen and NoteGen is required; the overall guarantee depends on how the private dataset is accessed by each module:
> - NoteGen is trained on the private training split under ($\epsilon_n, \delta_n$)-DP using the FastDP optimiser.
> - TermGen, when used with DP, operates on the private test split and satisfies ($\epsilon_t, \delta_t$)-DP.
>
> Because the two modules access disjoint subsets of the private dataset, their DP guarantees compose in parallel, and the overall privacy guarantee is simply the maximum of the two values. This is formalised in Equation (4), with a proof provided in Appendix C under the Term2Note settings. No additional advanced accounting is needed here because the privacy events do not overlap across modules.
>
> ***Weakness 3: inconsistent empirical results with the expected DP privacy-utility trade-off***
>
> Thank you for raising this important point. The observations in Table 1 may appear counterintuitive if one expects a strictly monotonic DP-utility trade-off. However, several factors help explain why a DP model may occasionally match or even exceed the utility of a non-DP model in our experiments. The behaviour appears only in the utility metrics. These metrics, F1, AUC, and Precision@k, capture different aspects of model performance. F1 reflects threshold-dependent classification accuracy, AUC measures ranking ability across all thresholds, and Precision@k evaluates correctness among the top-k predictions. Because these metrics emphasise different behaviours, it is expected that improvements may appear stronger on some metrics than others, especially given the difficulty of the downstream task (a 20-label classification problem) and the limited amount of training data (< 200 samples). For instance, even the original data shows a lower Precision@3 score than some synthetic datasets, indicating the challenge of the task rather than issues in the DP implementation.
>
>
> ***Weakness 4: unclear justification of the proposed framework***
>
> The key benefit of introducing a term-extraction stage is that it removes the common assumption in prior DP text-generation work that synthetic text must be conditioned on predefined class labels or global class distributions, which are often unavailable or require prior knowledge about the private corpus. By conditioning note generation on clinical terms, which can be obtained from any public clinical resources (not necessarily clinical notes), Term2Note enables flexible and class-agnostic synthesis of clinically flexible notes. Our experiments show that notes conditioned on these terms remain high-quality and clinically coherent.
>
> If the reviewer's concern refers specifically to the term generation module, its role is not to improve generation quality but to provide additional privacy protection by applying DP when reconstructing clinical terms from the private note. The synthetic notes produced with this extra protection are comparable in quality to those conditioned on ground-truth terms, demonstrating that this added layer of privacy does not degrade utility.

---

> ### Author Response · Authors · 2025-11-26
> **Response to Reviewer RmfR - Part 2**
>
> ***Weakness 5: justification of the baseline selection***
>
> Generating long-form clinical notes is known to be challenging, and many existing clinical text-generation models are not directly adaptable to our setting. For example, those synthetic clinical notes generated for the work Asclepius are conditioned on case reports from the PMC-Patients dataset, which is not available in our case. For this reason, we selected baselines that operate under comparable constraints, such as ICD-conditioned generation where ICD codes are available.
>
> Besides, our aim is not to evaluate FastDP itself. FastDP serves as a computationally efficient instantiation of DP-SGD within our framework, and any DP training method with equivalent guarantees could be substituted. The focus of our evaluation is therefore on the differences between Term2Note and alternative approaches to DP clinical note synthesis, rather than on benchmarking specific optimisers.
>
>
> ***Weakness 6: diagram lacks clarity***
>
> Thank you for the suggestion. In the revised submission, we will update the figure to clarify and better show where DP mechanisms are applied in Term2Note.
>
>
> ***Weakness 7: KL divergence of text length distribution provides limited insight***
>
> Firstly, we would like to clarify that we do not interpret KL divergence as a semantic similarity metric. Its purpose in our evaluation is narrowly defined: to quantify whether the distribution of lengths in the synthetic notes deviates from that of the real notes. By definition, DP mechanisms intentionally break one-to-one correspondence between real and synthetic instances. Consequently, pairwise structural or sentence-level comparisons are neither meaningful nor fully compatible with DP guarantees. Length-based distributional analysis therefore provides a simple, privacy-safe diagnostic that does not assume such correspondences.
>
> Secondly, we fully agree that text-length KL divergence provides only a narrow perspective on note quality. For this reason, we intentionally complement it with multiple additional evaluations, including clinical term distribution alignment and MAUVE for distributional similarity. As no single metric fully captures the structural and semantic properties of long clinical notes, our evaluation relies on a diverse set of measures to provide a more holistic assessment. We appreciate the suggestion and will consider incorporating additional structural or semantic metrics where appropriate in future work.

---

### Note · Authors · 2026-01-05

I have read and agree with the venue's withdrawal policy on behalf of myself and my co-authors.